# Improving Neural Program Induction by Reflecting on Failures

## Abstract

A neural program induction model is considered *good* if it is capable of learning programs with four objectives: (1) higher data efficiency, (2) a more efficient training process, (3) better performance, and (4) generalization for large-scale tasks. However, the neural program induction/synthesis models suffer from requiring a large amount of training iterations and training examples for training. Besides, the current state-of-the-art neural program induction models are still far from perfect in terms of performance and generalization when dealing with tasks that require complex task-solving logic. To mitigate these challenges, in this work, we present a novel framework called FRGR (Failure Reflection Guided Regularizer). Our proposed framework dynamically summarizes error patterns from the model's previous behavior and actively constrains the model from repeating mistakes of such patterns during training. In this way, the model is expected to converge faster and more data-efficiently as well as being less likely to fall into local optimum by making fewer mistakes of similar patterns. We evaluate FRGR based on multiple relational reasoning and decision-making tasks under both the data-rich and data-scarce settings. Experimental results show the effectiveness of FRGR in improving training efficiency, performance, generalization as well as data efficiency.

## 1 Introduction

> *"Failure is instructive. The person who really thinks learns quite as much from his failures as from his successes."*

—*John Dewey*

Program induction and synthesis has revolutionized many areas, e.g., inductive logic programming (ILP) (Evans & Grefenstette, 2018; Raghothaman et al., 2019; Rocktäschel & Riedel, 2017), software testing (Schilling & Müller, 2022; Kitzelmann, 2010; Cao et al., 2021), robotics (Polydoros & Nalpantidis, 2017), reinforcement learning (Jiang & Luo, 2019; Yang et al., 2021; Cao et al., 2022; Sun et al., 2020), etc. Recently, many deep learning-based program induction/synthesis models have been proposed (Chen et al., 2018; Devlin et al., 2017; Yin & Neubig, 2017). These neural (statistical) program induction/synthesis models are much more fault-tolerant and robust than traditional SAT-based synthesizers (Albarghouthi et al., 2017), which makes them more practical when dealing with noisy data. Besides, unlike traditional synthesizers which require symbolic input, the statistical models can also be trained directly on fuzzy input, such as raw images (Evans & Grefenstette, 2018), which makes them much more straightforward to use. However, the neural program induction/synthesis models suffer from two major challenges. First, they generally require a large amount of training iterations and training examples for training. Second, the current state-of-the-art neural program induction/synthesis models are still far from perfect in terms of performance and generalization when dealing with tasks that require complex task-solving logic.

Recently, provenance information (Cheney et al., 2009; Green et al., 2007; Cropper & Morel, 2021) is utilized to improve the efficiency of the SAT-based synthesizer (Raghothaman et al., 2019). Here, the provenance information refers to the reasoning process of a conclusion (e.g., the derivation trees in Figure 1(b)). When a wrong conclusion is derived, its provenance information can be used to guide the SAT solver in the following iterations to further reduce the search space and therefore to boost the training efficiency.

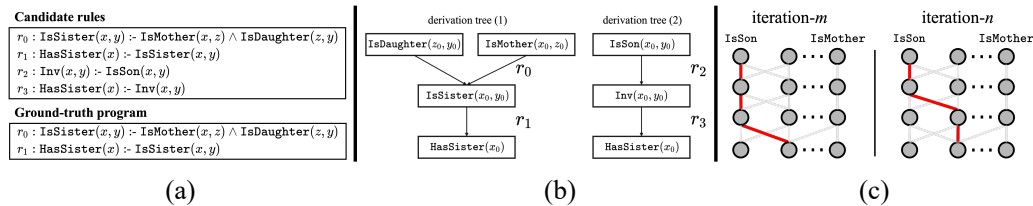

Figure 1: (a) An example candidate rules set and the desired ground-truth program of the task `HasSister`; (b) derivation trees that correspond to the application of the desired program (left) and an erroneous program (right); (c) example erroneous behaviors of a neural program induction model.

To make this concrete, Figure 1(a) shows an example search space of the candidate rules and the desired ground-truth program of the task `HasSister`. A desired program of this task should satisfy that: given $x$, `HasSister`$(x)$ is True when $x$ has at least one sister. Figure 1(b)(left) shows the derivation tree (provenance) of the ground-truth program $\{r_0, r_1\}$: first, the rule $r_0$ denotes that if $z$ is the mother of $x$ (`IsMother`$(x, z)$), and $y$ is the daughter of $z$ (`IsDaughter`$(z, y)$), then the conclusion that $y$ is the sister of $x$ (`IsSister`$(x, y)$) is deduced. Afterward, the rule $r_1$ deduces that $x$ has at least one sister if $y$ is its sister (`HasSister`$(x, y)$). Figure 1(b)(right) shows the derivation tree of an erroneous solution $\{r_2, r_3\}$. `Inv(x,y)` denotes an auxiliary predicate. To be concise, this erroneous program specifies that if $y$ is the son of $x$ (`IsSon`$(x, y)$), then $x$ has at least one sister. Clearly, if the derivation process involves the predicate `IsSon`, the solution would be either wrong or redundant. The derivation tree of the wrongly deduced conclusion is leveraged by (Raghothaman et al., 2019) (provenance) as constraints for the SAT solver. In this way, the SAT solver will not repeat the wrong reasoning process of similar patterns and thus improve the running efficiency.

Therefore, inspired by the effectiveness of provenance information for the SAT-based synthesizers, we propose a novel framework called FRGR (Failure Reflection Guided Regularizer)[1] to mitigate the above-mentioned two challenges of the neural program induction/synthesis models, namely, improving training efficiency and data efficiency, as well as performance and generalization. The main idea of FRGR is to dynamically summarize error patterns from the model's previous behavior and actively constrain the model from repeating mistakes of such patterns during training. As a concrete example, Figure 1(c) shows two instances of the erroneous reasoning process (behaviors) of a neural program induction model that aims to learn an implicit program for the `HasSister` task. Each neuron within the model semantically represents a predicate. The two examples denote the behavioral representation of the neural program induction model at iteration-$m$ and iteration-$n$, the red edge denotes the most activated rule for the output predicate during the reasoning process (we only show the activated weights of some predicates for better illustration). Similar to the erroneous derivation tree of the SAT-based synthesizer shown in Figure 1(b)(right), the behaviors of the neural induction model at the two different iterations both attribute high weights to the neuron which represents `IsSon`. Such behaviors would result in the model's erroneous conclusions. FRGR extracts and utilizes such behaviors to refrain the model from repeating wrong behaviors.

The detailed workflow of FRGR is as follows. Firstly, FRGR dynamically extracts and stores the behaviors of the neural induction model during training iterations. Secondly, FRGR applies the pattern mining algorithm (Agrawal & Srikant, 1994) on the historical behaviors and summarizes the error patterns. Lastly, FRGR regularizes the model for making similar mistakes with the behavioral loss which is derived based on the matching of the model's current behavior and the derived error patterns. By doing so, the model is constrained from making similar mistakes. And it is expected to converge faster and more data-efficiently as well as being less likely to fall into local optimum by making fewer mistakes of similar patterns.

Our contributions are threefold: ❶ We propose a novel framework for the neural program induction model, called FRGR. The idea is to dynamically summarize the error patterns based on pattern mining on its historical behaviors, and then regularize the model if it repeats similar error patterns. ❷ Under the data-rich setting, extensive experimental results on multiple well-known relational reasoning and decision-making benchmarks validate that FRGR can considerably improve the training efficiency, and improve the performance and generalization of the neural program induction model. ❸ We demonstrate that FRGR can considerably boost the induction models' data efficiency. Specifically, our model significantly outperforms the baseline when the training data volume is extremely scarce and our method can generally help achieve (near-)optimal performance with much fewer training examples.

---

[1] The implementation is available at: `https://anonymous.4open.science/r/FRGR-C763`

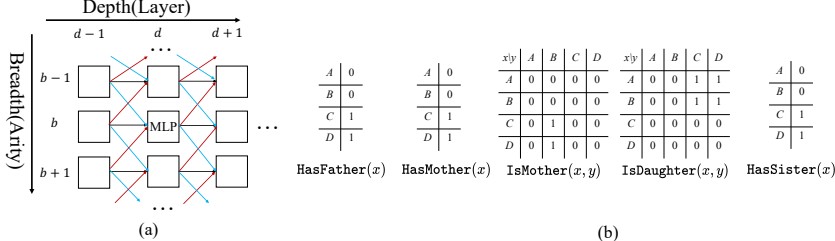

Figure 2: (a) High-level architecture of the NLM model. (b) Truth value table representation of the `HasSister` task sample. Given the truth values of the input predicates `HasMother`, `HasFather` `IsMother` and `IsDaughter`, the NLM model is required to deduce the truth value representations of the target predicate `HasSister`.

## 2 PRELIMINARY

In this section, we introduce the fundamentals of inductive logic programming (ILP) which serves as the bedrock of neural logic program induction/synthesis. We also illustrate the basics of neural logic machines (NLM), which is the backbone neural program induction model we study in this work.

### 2.1 INDUCTIVE LOGIC PROGRAMMING

Logic programming (LP) is a programming paradigm based on first-order logic (FOL). Predicate and term serve as the two most fundamental elements of FOL. A predicate $p$ can denote attributes of an object or relations among objects. An atom denotes a predicate $p$ followed by a tuple $p(x_1, \ldots, x_n)$, where $p$ is a $n$-ary predicate and $x_1, \ldots, x_n$ are called terms, which can be either variables or constants. An atom is grounded if all terms in the atom are constants. A clause (rule) is denoted as $H \leftarrow B_1 \wedge \ldots \wedge B_n$, where $H$ is called the head atom, $B_1 \wedge \ldots \wedge B_n$ is called the body. It is interpreted as: $H$ is true if the body $(B_1 \wedge B_2 \cdots \wedge B_n)$ is true. A logic program is a program that consists of a set of clauses that satisfies certain specifications, which can be represented by I/O examples (Evans & Grefenstette, 2018; Chen et al., 2019) or natural language (Gulwani & Marron, 2014; Manshadi et al., 2013; Desai et al., 2016)) Inductive Logic Programming (ILP) (Getoor & Taskar, 2007; Muggleton, 1991; Getoor et al., 2001) is in essence a searching task that aims to find a logic program. Given a set of background atoms provided in prior $\mathbf{B}$ (also called premise), a set of positive examples $\rho$, and a set of negative samples $\eta$, the goal of a program induction/synthesis model is to derive a logic program $C$ that satisfies the following two conditions: (1) based on the premise $\mathbf{B}$, $C$ entails all the positive examples, denoted by $\forall \rho : \mathbf{B}, C \models \rho$, and (2) based on the premises, $C$ does not entail any of the negative examples, denoted by $\forall \eta : \mathbf{B}, C \not\models \eta$.

### 2.2 NEURAL LOGIC MACHINES

The neural logic machines (NLM) model (Dong et al., 2018) is a neural-symbolic network architecture with strong inductive bias that realizes the logic machines in a neural network manner. The model simulates the forward chaining mechanism by approximating the logic operations with neural networks. The NLM model architecture is shown in Figure 2(a). Concretely, it consists of NLM block that implements the *Boolean logic rules* and *logic quantification*. The *breadth* denotes input of predicates from different arities. And by stacking layers of multiple *depth*, the NLM model is capable of implementing the forward chaining mechanism.

The NLM neural program induction model uses the probabilistic tensor representation for logic predicates. Given an object set $\mathcal{U}$, where $|\mathcal{U}| = m$. The object set is represented with truth-value matrices that denote the instantiation of different $n-$ary basic input predicates based on the object set. For instance, Figure 2(b) shows the concrete truth value representations of the input & output predicates of the `HasSister` task sample. Concretely, it represents a family with four members $U = \{A, B, C, D\}$. $A$ is the mother, $B$ is the father, $C$ and $D$ are the two daughters of them. The input representation contains the truth value matrices of the input basic predicate `HasMother(x)`, `HasFather(x)`, `IsMother(x,y)`, `IsDaughter(x,y)`. *E.g.*, for the `IsMother(x,y)`, since $B$ is the mother of both $C$ and $D$, the items within the matrix that represent `IsMother(C,B)` and `IsMother(D,B)` equals to 1. Then given the input representation, the models are required to deduce the truth values of the desired target predicate `HasSister`. Then the truth-value matrices in categorized according to their arities and are used as input for the respective NLM block located at the different breadth. Concretely, for the matrices of Figure 2(b), the representations of the unary predicates `HasMother(x)`, `HasFather(x)` is used as input for the breadth=1 NLM block, and the

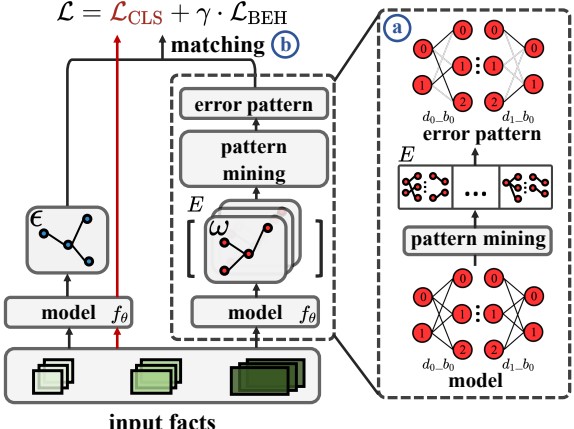

Figure 3: Overview of the FRGR framework. (a) during the error pattern mining phase, the erroneous behavior list $E$ stores the behavior representations of historical model checkpoints with erroneous conclusions. Pattern mining is then conducted based on the erroneous behavior list and it dynamically summarizes error patterns from historical iterations. (b) During the behavioral regularization phase, the behavior representation of the current neural program induction model $\omega$ is matched against the error pattern $\epsilon$, based upon which the behavioral loss is derived.

representations of the binary predicates $\texttt{IsMother}(x, y)$, $\texttt{IsDaughter}(x, y)$ is used as input for the breadth=2 NLM block.

Then given the truth-value represented predicates with arities ranging from $[0, B]$, each NLM block proposes a novel neural-symbolic model architecture which approximates the *Boolean logic rules* and *logic quantification*. First, for the *Boolean logic rules* implementation, the model approximates the Horn clause: $\hat{p}(x) \leftarrow p_1(x) \wedge \cdots \wedge p_g(x)$, where $\hat{p}(x)$ denotes the output conclusive predicate and $p_k(x), k \in [1, \ldots, g]$ denotes $g$ different input predicates. To achieve this, NLM block introduces using the multi-layer perceptron (MLP) module, and applies it on the representations of different predicates, formally: $\hat{p}(x_1, \cdots, x_n) = \sigma(\text{MLP}(p_1(x_1, \cdots, x_n), \cdots, p_k(x_1, \cdots, x_n))$ where $p_k, k \in [1, \ldots, g]$ denotes $g$ different predicates, $\hat{p}$ is the generated conclusive predicate, $\sigma$ denotes the sigmoid function. Intuitively, the MLP serves as the input predicate selection and conjunction module. In this way, the rule $\texttt{HasSister}(x) \leftarrow \texttt{IsDaughter}(z, y) \wedge \texttt{IsMother}(x, z)$ can be approximated via this implementation which denotes the semantics that $x$ has sister if $z$ is $x$'s mother and $y$ is $z$'s daughter.

Next, for the *logic quantification*, this mechanism allows the NLM block to leverage predicates of different arities for deduction and realize the $\forall$ and $\exists$ quantifiers. Specifically, the input $b$-ary predicates of each NLM block are derived from three channels from the previous layer: $b-1$-ary predicates using the Expansion operation, $b+1$-ary predicates using the Reduction operation, and $b$-ary predicates from the same breadth (i.e., $P_d^b = \{\text{Expansion}(P_{d-1}^{b-1}) \cup P_{d-1}^b \cup \text{Reduction}(P_{d-1}^{b+1})\}$). The Expansion and Reduction operations represent two types of meta-rules for realizing quantification. Specifically, the Expansion operation (denoted by blue arrows in Figure 2) denotes the meta-rule: $\forall x_{b+1}\ q(x_1, x_2, \cdots, x_b, x_{b+1}) \leftarrow p(x_1, x_2, \cdots, x_b)$, where $q$ denotes a predicate derived from predicate $p$ by introducing a new variable $x_{b+1}$. Similarly, the Reduction operation (denoted by red arrows in Figure 2(a)) denotes the meta-rule: $q(x_1, x_2, \cdots, x_b) \leftarrow \forall x_{b+1}\ p(x_1, x_2, \cdots, x_b, x_{b+1})$, where $q$ denotes a predicate derived by reducing a variable in predicate $p$ via the quantifier.[2].

## 3 METHODOLOGY

In this section, we introduce our novel regularization mechanism, called FRGR (Failure Reflection Guided Regularizer) to improve the training and data efficiency as well as the performance and generalization of the neural program induction model. FRGR consists of two key phases: ❶ error pattern mining and ❷ behavioral regularization, as shown in Figure 3. In phase ❶, FRGR dynamically summarizes error patterns from the historical reasoning process (also called behavior in this work) of the model. In phase ❷, FRGR constrains the model's behavior by penalizing it for repeating the historical error pattern. The details of each phase are introduced in the following subsections.

---

[2]Please refer to Appendix D for more details about the Expansion, Reduction and Permute operations.

## 3.1 Error Pattern Mining

In the error pattern mining phase, FRGR dynamically summarizes the neural program induction model's historical error patterns during its training iterations, as shown in Figure 3(a). There are two key components involved in this phase: (1) erroneous behavior list $E$ and (2) error pattern $\epsilon$. Algorithm 1 (lines 1–12) shows the detailed steps of this phase [3].

Given a set of input predicates grounded on objects $\mathbf{X}$, i.e., the background facts (premises), the neural program induction model infers the conclusions $\tilde{\mathbf{Y}}$ by performing deduction based on first-order logic (Pearl et al., 2000), as shown in line 4 of Algorithm 1. During the training iterations, if an error conclusion is deduced, FRGR extracts the current behavior representation $\omega$ of the model checkpoint (i.e., the representation of the model's reasoning process in the current iteration), as shown in line 5 of Algorithm 1, where $f_\theta(\cdot)$ denotes the feed-forward process of a neural program induction model and $\rho(\cdot)$ denotes the extraction process. Specifically, for the NLM model we study in this work, for each multi-layer perceptron (MLP) of all the computational units, we consider the largest weight associated with the output conclusive predicate (i.e., the associated weight of the predicate that contributes most to the corresponding output logic expression):

$$\rho(f_\theta) = \{c \parallel v \mid v = (id(o, O^o), \arg\max_j \theta^o_{I,c}[j]), \ c \in D \times B\}, \tag{1}$$

where $\parallel$ denotes the tuple concatenation operation, $D = \{1, 2, \ldots, d\}$; $B = \{1, 2, \ldots, b\}$; $d$ and $b$ denote the number of depth and breadth of the NLM model architecture respectively; $c$ is the index of each computation unit; $I$ and $O$ represent the set of input and output neurons (predicates) of each computational unit; $\theta$ represents the model's learnable weights. $id(\cdot)$ is a function that identifies the index of the neuron $o$ in its corresponding output layer $O^o$. E.g., $(d_1, b_0, o_1, i_2)$ denotes the weight that is located at the computational unit at depth 1 and breadth 0: $(d_1, b_0)$, and it associates the first output predicate and the second input predicate: $(o_1, i_2)$. The behavior representation $\omega$ therefore stores the coordinates of the selected weights (i.e., the identifier of the predicates). Then FRGR uses $\omega$ to update the erroneous behavior list $E$ dynamically, as shown in lines 6–9, where $E \in \mathbb{R}^\tau$, $\tau$ denotes the size of the list.

After obtaining the erroneous behavior list $E$, to avoid penalizing the wrong weights that overlap with the correct behavior, FRGR applies the Apriori frequent item set mining algorithm (Agrawal & Srikant, 1994) on $E$ to get the error pattern (lines 10–12). With this step, FRGR is able to identify the coexistence of the predicates among all historical behavior representations that derive erroneous conclusions as the error pattern $\epsilon$.

Figure 3 part(a) shows a concrete example of error pattern mining. The number in each neuron denotes its index within its corresponding layer (view vertically). The symbol $d_x\_b_y$ indicates the positional index of the computational unit at depth $d$ and breadth $b$ within the model architecture. For the two presented computational units $d_0\_b_0$ and $d_1\_b_0$ in the figure, FRGR extracts their behavior representation in multiple iterations with $\rho$ and stores them in $E$. To make this concrete, let's consider the erroneous behaviors of HasSister task in Figure 1(c). If the model erroneously inferred a solution that relies heavily on the predicate IsSon and the neuron that denotes the invented predicate HasSon$(x)$:-IsSon$(x, y)$, HasSister$(x)$:-HasSon$(x, y)$, their corresponding weights will be extracted by error pattern mining via Equation (1). More specifically, the output of $\rho(f_\theta)$ consists of the indices of corresponding weights that are associated with these predicates. The index of the corresponding weight of HasSon is: $(d_0, b_2, o_1, i_2)$, $o_1$ means that HasSon is the first output predicate of this NLM block, $i_2$ means that IsSon is the second input predicate of this NLM block. Similarly, the index of the corresponding weight of HasSister is: $(d_1, b_1, o_2, i_2)$. Note that the position of input and output predicates in each NLM block are arbitrarily assigned for illustration. Then, the set $\{(d_0, b_2, o_1, i_2), (d_1, b_1, o_2, i_2)\}$ is the current behavior representation $\omega$, which is used by FRGR to update the erroneous behavior list $E$. After the mining process, FRGR manages to identify the error pattern representation that stores the position index of the frequently intersected weights of all historical erroneous behavior representations: $\epsilon = [(d_0, b_0, o_0, i_0), \ (d_0, b_0, o_2, i_0), \ \ldots, \ (d_1, b_0, o_0, i_0)]$. The error pattern representation is then used for the following behavioral regularization.

---

[3]We use the implementation of our framework on the supervised learning tasks for illustration. Please refer to Appendix A for the detailed implementation of our method on the reinforcement learning tasks.

---

**Algorithm 1:** FRGR regularization framework.

---

**Input:** Training mini-batches $\mathcal{B}$, Neural induction model $f$, Erroneous behavior list $E$, Erroneous behavior list size $\tau$, Error pattern $\epsilon$, Behavior extractor $\rho$, Regulatory coefficient $\gamma$, Learning rate $\eta$, Model weights $\theta$, Classification loss $\mathcal{L}_{\text{CLS}}$, Behavioral loss $\mathcal{L}_{\text{BEH}}$

```
/* Error Pattern Mining */
```
1  Initialization: $i \leftarrow 0$
2  **for** $mini\_batch \in \mathcal{B}$ **do**
3  $\quad$ $\mathbf{X}, \mathbf{Y} \leftarrow mini\_batch$
4  $\quad$ $\tilde{\mathbf{Y}} \leftarrow f_\theta(\mathbf{X})$ $\qquad\qquad$ `// deduce conclusions given input facts`
5  $\quad$ $\omega \leftarrow \rho(f_\theta(\cdot))$ `// extract behavior representation of current model`
6  $\quad$ **if** $\mathbf{Y} != \tilde{\mathbf{Y}}$ **then**
7  $\quad\quad$ $E[i\%\tau] \leftarrow \omega$ $\qquad\qquad$ `// update erroneous behavior list`
8  $\quad\quad$ $i \leftarrow i + 1$
9  $\quad$ **end**
10 $\quad$ **if** $i \% \tau == 0$ **then**
11 $\quad\quad$ $\epsilon \leftarrow Apriori(E)$
12 $\quad$ **end**
```
   /* Update with behavioral regularization */
```
13 $\quad$ $\mu \leftarrow \omega \cap \epsilon$ $\qquad\qquad\qquad$ `// erroneous pattern matching`
14 $\quad$ $\mathcal{L}_{\text{BEH}} \leftarrow \gamma \cdot \sum_{\nu \in \mu} ||\theta[\nu]||_1$
15 $\quad$ $\mathcal{L} \leftarrow \mathcal{L}_{\text{CLS}}(\mathbf{Y}, \tilde{\mathbf{Y}}) + \gamma \cdot \mathcal{L}_{\text{BEH}}$
16 $\quad$ $\theta \leftarrow \theta - \eta \nabla_\Theta \mathcal{L}$
17 **end**

---

## 3.2 BEHAVIORAL REGULARIZATION.

With the error pattern $\epsilon$ obtained in the error pattern mining phase, FGFR utilizes it together with the model's current behavior representation to perform the behavioral regularization. In this way, the model is expected to refrain from repeating similar errors depicted by the error pattern. Therefore, it is more likely to find a desirable solution instead of converging to a local optimum. The detailed steps of the behavioral regularization phase are shown in lines 13–17 of Algorithm 1.

Specifically, FGFR performs matching between the error pattern and the current behavior representation. In this work, we consider matching by calculating the intersection between the error pattern set $\epsilon$ and the model's current behavior representation $\omega$, as shown in line 13. This matching factor, denoted by $\mu$, measures the degree of repetitions (regarding the error pattern presented previously) performed by the model. Based on $\mu$, the behavioral regularization and the final training loss are calculated as follows:

$$\begin{aligned} \mathcal{L}_{\text{final}} &= \mathcal{L}_{\text{CLS}}(\mathbf{Y}, \tilde{\mathbf{Y}}) + \gamma \cdot \mathcal{L}_{\text{BEH}} \\ &= \mathcal{L}_{\text{CLS}}(\mathbf{Y}, \tilde{\mathbf{Y}}) + \gamma \cdot \sum_{\nu \in \mu} ||\theta[\nu]||_1, \ where \ \mu = \omega \cap \epsilon \end{aligned} \quad (2)$$

where $\mathcal{L}_{\text{CLS}}$ denotes the common classification loss, and $\gamma$ denotes the regulatory coefficient for the behavioral loss $\mathcal{L}_{\text{BEH}}$, $\theta$ denotes the learnable weight tensor of the neural program induction model. To calculate the behavioral regularization term, we obtain the positional indices of the intersected weights set of the error pattern representation $\epsilon$ and the model's current behavior representation $\omega$: $\mu = \omega \cap \epsilon$ and apply the L1 norm to aggregate the selected weights: $\sum_{\nu \in \mu} ||\theta[\nu]||_1$, as shown in line 14 of Algorithm 1. The behavioral regularization $\mathcal{L}_{\text{BEH}}$ is then used along with the classification loss $\mathcal{L}_{\text{CLS}}$ to update the model. By regularizing the neural program induction model in this way, the model is refrained from repeating similar mistakes during the learning process and is able to achieve more efficient induction and converge to the global optima.

## 4 EXPERIMENTS

In this section, we evaluate the effectiveness of the FRGR framework. We implemented our FRGR on the foundation of previous state-of-the-art neural program induction model NLM under two scenarios: ❶ the ideal data-rich setting, and ❷ the data-scarce setting (simulation of real-world limited training data scenario). We conduct experiments on two representative categories of benchmarks that are adopted by previous works (Dong et al., 2018; Zimmer et al., 2021): relational reasoning tasks and reinforcement learning (RL) tasks.

Table 1: The comparative results with original NLM model and NLM w/ FRGR for family tree reasoning, general graph reasoning, and reinforcement learning. $n$ is the size of the task, e.g., how many members in a family. For RL tasks, all models are trained using curriculum learning on environments with $n \leq 12$.

| Task | NLM | | | | NLM w/ FRGR (Ours) | | | | # Examples/ |
|---|---|---|---|---|---|---|---|---|---|
| Family Tree | Grad-ratio | n=20 | n=100 | # Iterations | Grad-ratio | n=20 | n=100 | # Iterations | Episodes |
| HasFather | 100.00% | 100.00%±0.00 | 100.00%±0.00 | 5.90±2.23 | 100.00% | 100.00%±0.00 | 100.00%±0.00 | 6.00±2.49 | 50,000 |
| HasSister | 100.00% | 100.00%±0.00 | 100.00%±0.00 | 18.09±7.98 | 100.00% | 100.00%±0.00 | 100.00%±0.00 | 17.64±5.68 | 50,000 |
| IsGrandparent | 100.00% | 100.00%±0.00 | 100.00%±0.00 | 96.20±21.87 | 100.00% | 100.00%±0.00 | 100.00%±0.00 | 55.80±5.57 | 100,000 |
| IsUncle | 90.00% | 99.76%±0.01 | 82.60%±0.31 | 143.70±81.43 | 100.00% | 100.00%±0.00 | 100.00%±0.00 | 82.50±17.60 | 100,000 |
| IsMGUncle | 70.00% | 97.16%±0.07 | 10.04%±0.21 | 203.88±97.16 | 100.00% | 99.96±0.001 | 60.44%±0.40 | 175.20±61.38 | 200,000 |
| Graph Reasoning | Grad-ratio | n=10 | n=20 | # Iterations | Grad-ratio | n=10 | n=20 | # Iterations | |
| 1-OutDegree | 100.00% | 100.00%±0.00 | 100.00%±0.00 | 14.30±7.45 | 100.00% | 100.00%±0.00 | 100.00%±0.00 | 17.00±9.06 | 50,000 |
| 2-OutDegree | 90.00% | 96.52%±0.10 | 90.80%±0.28 | 77.9±150.5 | 100.00% | 100.00%±0.00 | 100.00%±0.00 | 13.40±11.13 | 100,000 |
| 4-Connectivity | 100.00% | 100.00%±0.00 | 100.00%±0.00 | 16.80±4.61 | 100.00% | 100.00%±0.00 | 100.00%±0.00 | 20.50±7.98 | 50,000 |
| 6-Connectivity | 40.00% | 64.96%±0.39 | 53.28%±0.34 | 339.9±211.45 | 90.00% | 97.12%±0.09 | 94.60%±0.16 | 69.20±153.32 | 50,000 |
| Reinforcement Learning | Grad-ratio | n=10 | n=50 | # Iterations | Grad-ratio | n=10 | n=50 | # Iterations | |
| Sorting | 100.00% | 100.00%±0.00 | 100.00%±0.00 | 24.00±8.49 | 100.00% | 100.00%±0.00 | 100.00%±0.00 | 22.20±4.05 | 1,000 |
| Path | 50.00% | 99.55%±0.01 | 99.95%±0.001 | 311.00±107.15 | 60.00% | 100.00%±0.00 | 100.00%±0.00 | 305.20±109.97 | 24,000 |
| BlocksWorld | 40.00% | 97.11%±0.060 | 76.89%±0.34 | 390.11±95.05 | 60.00% | 96.59%±0.057 | 83.90%±0.28 | 386.67±132.75 | 50,000 |

## 4.1 BENCHMARKS AND SETUP

**Relational reasoning tasks.** The relational reasoning tasks contains two major categories: Family Tree Reasoning (Dong et al., 2018; Evans & Grefenstette, 2018) and General Graph Reasoning (Graves et al., 2016; Dong et al., 2018; Zimmer et al., 2021). For the Family Tree Reasoning tasks, a family tree sample consists of $n$ family members. The goal of this task is to learn the properties of family members or the relations between them: HasFather, IsUncle, etc. For the General Graph Reasoning tasks, an undirected graph sample consists of $n$ nodes. The relationships between nodes are represented by the predicate HasEdge. The goal of this task is to acquire the properties of the nodes or the relations between them: $k$-Connectivity and $k$-OutDegree. For $k$-Connectivity, the model needs to determine if two nodes can be connected by a path with at most $k$ edges; for $k$-OutDegree, it needs to classify if the out-degree of the node equals to $k$.

**Reinforcement learning tasks.** We evaluate FRGR on three RL tasks: Sorting, Path (Graves et al., 2016), and Blocks World (Nilsson, 1982; Gupta & Nau, 1992). For the Sorting task, an array of length $m$ is used as input, and the goal is to learn the swap predicate (i.e., swapping two integers in the array) to sort the list in ascending order. For the Path environment, given an undirected graph, the goal is to find a path between the start node $s$ and the end node $e$, which are represented by two unary predicates (IsStart, IsEnd).

The Blocks World task includes two worlds: an initial world and a target world, both of which contain $m$ objects ($m - 1$ cubes and 1 ground). The goal is to learn how to move the objects to change the world from the initial setting to the target setting[4].

**Training Setup.** For the Family Tree Reasoning task, all the models are trained on family trees with 20 family members and tested on family sizes of 20 and 100. For the General Graph Reasoning task, all the models are trained on graphs with 10 nodes and tested on graphs with 10 and 20 nodes. For RL tasks, all the models are trained with curriculum learning (Bengio et al., 2009) via REINFORCE algorithm (Williams, 1992) using environments containing less than 12 objects and tested on 10 and 50 objects. Please refer to Appendix B for detailed training settings.

**Evaluation Metrics.** We adopt the evaluation metrics used in previous works (Dong et al., 2018; Zimmer et al., 2021). The *success rate* is used to evaluate IID test performance and OOD (out-of-distribution) generalization to large-scale tasks. It is defined as the ratio of achieving $100\%$ accuracy of the task[5]. *Graduation ratio* measures the percentage of the training instances of different seeds that reach a success rate of $100\%$. Finally, training iteration measures the number of training epochs required to train the model till reaching optimal success rate on the validation set. We use it to measure training efficiency.

---

[4]Please refer to Appendix C for more details regarding benchmarks.

[5]For example, given a test set of the IsUncle task, containing 300 families and each contains 20 family members, the success rate refers to the ratio of the families that achieve $100\%$ accuracy (e.g., the success rate is $10\%$ if the model gives the right conclusion for every pair of the family members in 30 out of 300 families).

Table 2: The comparison between NLM and NLM w/ FRGR when training data is extremely scarce. The training settings are the same as the data-rich scenario except for the number of training data.

| Task | NLM | | | | NLM w/ FRGR (Ours) | | | | # Examples/ Episodes |
|---|---|---|---|---|---|---|---|---|---|
| Family Tree | Grad-ratio | n=20 | n=100 | # Iterations | Grad-ratio | n=20 | n=100 | # Iterations | |
| HasFather | 100.00% | 100.00%±0.00 | 100.00%±0.00 | 5.50±2.76 | 100.00% | 100.00%±0.00 | 100.00%±0.00 | 5.50±2.76 | 100 |
| HasSister | 100.00% | 100.00%±0.00 | 100.00%±0.00 | 13.20±2.49 | 100.00% | 100.00%±0.00 | 100.00%±0.00 | 13.30±2.58 | 100 |
| IsGrandparent | 90.00% | 68.70%±0.44 | 63.40%±0.46 | 127.90±132.59 | 100.00% | 77.51%±0.41 | 72.26%±0.42 | 54.30±6.03 | 200 |
| IsUncle | 100.00% | 96.49%±0.06 | 62.53%±0.46 | 134.30±51.41 | 100.00% | 98.05%±0.06 | 81.50%±0.37 | 102.8±44.59 | 200 |
| IsMGUncle | 70.00% | 68.52%±0.33 | 33.00%±0.43 | 356.3±125.52 | 100.00% | 94.20±0.08 | 48.00%±0.48 | 251.6±66.00 | 400 |
| Graph Reasoning | Grad-ratio | n=10 | n=20 | # Iterations | Grad-ratio | n=10 | n=20 | # Iterations | |
| 1-OutDegree | 100.00% | 100.00%±0.00 | 100.00%±0.00 | 52.90±29.97 | 100.00% | 100.00%±0.00 | 100.00%±0.00 | 61.10±32.49 | 100 |
| 2-OutDegree | 90.00% | 99.92%±0.002 | 99.72%±0.001 | 135.7±150.52 | 100.00% | 100.00%±0.00 | 100.00%±0.00 | 73.30±80.48 | 200 |
| 4-Connectivity | 100.00% | 100.00%±0.00 | 100.00%±0.00 | 151.6±101.30 | 100.00% | 100.00%±0.00 | 100.00%±0.00 | 195.10±98.01 | 100 |
| 6-Connectivity | 80.00% | 99.89%±0.002 | 99.57%±0.005 | 63.75±53.90 | 80.00% | 99.56%±0.01 | 99.81%±0.004 | 38.25±15.10 | 100 |
| Reinforcement Learning | Grad-ratio | n=10 | n=50 | # Iterations | Grad-ratio | n=10 | n=50 | # Iterations | |
| Sorting | 100.00% | 100.00%±0.00 | 100.00%±0.00 | 28.20±8.97 | 100.00% | 100.00%±0.00 | 100.00%±0.00 | 24.60±5.97 | 140 |
| Path | 70.00% | 98.94%±0.01 | 93.65%±0.10 | 304.60±155.77 | 100.00% | 99.88%±0.002 | 99.80%±0.006 | 206.00±82.45 | 600 |
| BlocksWorld | 40.00% | 84.13%±0.30 | 45.93%±0.45 | 414.00±133.10 | 40.00% | 90.13%±0.13 | 52.60%±0.37 | 442.00±81.76 | 1100 |

## 4.2 DATA-RICH SCENARIO

We follow the setup of previous works (Dong et al., 2018; Zimmer et al., 2021) and conduct experiments under the ideal data-rich setting (i.e., the number of training samples are unlimited) to demonstrate the usefulness of FRGR. The results are shown in Table 1.

First, concerning the graduation ratio, NLM w/ FRGR (Ours) achieves substantial improvements for both relational reasoning and reinforcement learning tasks. Notably, it surpasses the original NLM model by achieving optimal graduation ratios (100%) across all family tree reasoning tasks.

Second, in terms of IID test performance and OOD generalization of large-scale tasks, NLM w/ FRGR consistently outperforms the NLM model for the majority of cases.

Finally, for training efficiency, we observe that the model requires much fewer iterations to converge with FRGR in general. For the relational reasoning benchmarks, our method requires 42.0% fewer iterations than NLM to converge to the optimal solution on the IsGrandparent task; We attribute this to the fact that these tasks are rather simple and straightforward to learn (less than 20 iterations are required for NLM to find optimal solutions).

## 4.3 DATA-SCARCE SCENARIO

We evaluate FGFR under the data-scarce setting to see if it works in real-world scenarios where the training examples are usually limited. In the extreme data-scarce setting, we use only $1/500$ of the data-rich training data volume on the relational reasoning benchmark. The detailed training data volume and results are shown in Table 2. To further investigate the model's performance in relation to training data volume, we gradually increase the number of training examples. The results are shown in Figure 4, where we vary the number of training examples from 100 to 1000, all of which are significantly fewer than the data-rich scenario.

In the extreme data-scarce setting, the results in Table 2 clearly indicate that NLM w/ FRGR (ours) is effective in improving the baseline NLM model. Take the IsMGUncle task for example. When using only 400 training examples, our method has already achieved 100.00% graduation ratio, +25.68% in terms of performance, +15.00% in terms of generalization, and $-29.39\%$ fewer iterations compared to the original NLM. It is also noteworthy that the generalization is already considerably better than the NLM under the data-rich setting (10.04%).

Regarding the effect of the training data volume, we observe that our method generally outperforms NLM across different training data volumes in terms of training efficiency, performance, and generalization (shown in Figure 4). Additionally, our method works particularly well when the training data is extremely scarce. We notice that FRGR is effective in enabling the model to achieve optimal or near-optimal results with fewer training examples. Take the IsUncle task as an example, our method takes only 800 examples to reach optimal performance while NLM fails to do so even with $1,000$ examples. Regarding training efficiency, with the incorporation of FRGR, the model requires fewer iterations to converge to a desired program than NLM for all evaluated data volumes. Specifically, for the Path task, when using 800 training examples, NLM requires 243.8 iterations on average while ours only requires 190.4 iterations . We can therefore conclude from the above findings that FRGR is effective in boosting the data efficiency of the neural program induction model.

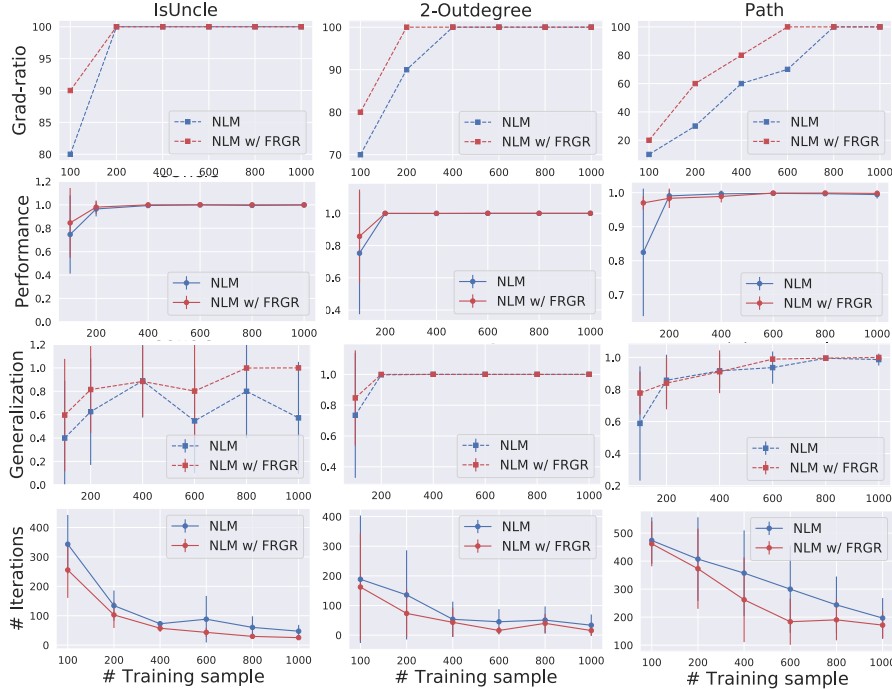

Figure 4: The training progress of NLM model and NLM w/ FRGR when the samples are scarce. The x-axis indicates the number of samples fed to the models during the training. All results are averaged over 10 random seeds.

## 5 RELATED WORK

**Neural Program Induction & Synthesis.** Program induction and synthesis are the tasks that aim to learn programs that satisfy pre-defined program specifications. In recent years, with the development of deep learning, neural networks-based program induction and synthesis have been proven effective in logical reasoning tasks with much less manual design effort and higher fault-tolerance (Evans & Grefenstette, 2018; Devlin et al., 2017; Chen et al., 2018; Bunel et al.). Evans et al. (Evans & Grefenstette, 2018) proposes a differentiable implementation of inductive logic programming ($\partial$ILP) which is capable of synthesizing white-box Datalog programs given noisy input data. Differentiable Logic Machines (DLM) (Zimmer et al., 2021) introduce a novel neural-logic architecture and training algorithm that helps generate fully-interpretable logic programs. Trivedi et al. (Trivedi et al., 2021) propose a neural program synthesis framework that first learns a program embedding space that parameterizes behaviors in an unsupervised manner, then generates a program that maximizes the return of a task by searching over the program embedding space. The proposed framework manages to outperform previous DRL and program synthesis baselines.

**Relational Inductive Bias and Provenance.** Relational inductive biases of neural network architectures can improve models' learning about entities and relations. For example, the relational inductive biases of graph networks can improve combinatorial generalization and sample efficiency (Battaglia et al., 2018). It has been found that query provenance (Cheney et al., 2009; Green et al., 2007) provides a useful mechanism for analysis and synthesis of Datalog programs, e.g., finding abstractions for program analysis written in Datalog (Zhang et al., 2014), and scaling up the synthesis process of Datalog programs based on the provenance information learned from SAT constraints solving to further reduce the size of the search space for program synthesis (Raghothaman et al., 2019).

## 6 DISCUSSIONS & CONCLUSIONS

In this work, we propose a novel regularization framework called FRGR, which aims to improve the training and data efficiency as well as the performance and generalization of the neural program induction models. Our proposed method first summarizes error patterns from the model's historical behavior with pattern mining techniques. Based on that, behavioral regularization is conducted by matching the current model behavior and the summarized error pattern. In this way, the model is constrained from repeating mistakes of such patterns during training. Experimental results on multiple relational reasoning and decision-making benchmarks demonstrate that our proposed framework can effectively improve training efficiency, performance, generalization as well as data efficiency.

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
