# Improving Neural Program Induction by Reflecting on Failures

## A  FRGR for the Reinforcement Learning Tasks

In this section, we illustrate the FRGR framework for reinforcement learning tasks. Reinforcement learning tasks focus on making sequential decisions to complete the tasks with the goal of maximizing the returns (at time step $t$, the parameterized policy will take action $a_t$ at state $s_t$, and continue until the task is completed or reaches the maximum steps allowed). Consequently, determining the correctness of a single action within an RL sequence is challenging. Therefore, after finishing a sequence, the final return of each state-action pair taken in that sequence will be calculated using a discounting factor. RL algorithms mainly optimize policies toward the direction of maximizing the returns. Similarly, in FRGR, the discounted returns are also used to determine which behavioral snapshots are used for error pattern mining. Algorithm 1 (line 1-8) shows the detailed steps.

Given a reinforcement learning task, the induction model (policy $\pi_\theta$) interacts with the environment for a maximum of $T$ steps, as shown in line 3. For each step $t$, the induction model takes the grounded state $s_t$ and outputs the action $a_t$. The behavioral snapshot is also extracted ($\omega_t$). Then, by interacting with the environment via $a_t$, the environment gives the reward $r_t$ and next state $s_{t+1}$, as shown in line 4. After one episode, the discounted returns $(G_1, G_2, \ldots, G_T)$ are calculated based on the rewards collected, as shown in line 5. If the induction model fails to complete the task in this episode, we update the erroneous behavior list $E$ based on the discounted returns. Specifically, $E$ is implemented as a max heap with the size $\tau$, storing the tuple $(G_t, \omega_t)$. For every new tuple $(G_i, \omega_i)$, when its return is smaller than the return of the root node, it is added to the erroneous behavior list, as shown from line 6 to line 8. In this way, the behavioral snapshots that cause the lowest returns are considered the (most) erroneous behaviors. Finally, similar to the relational reasoning scenario, the erroneous behavior list is used for error pattern mining, as shown in line 9. The summarized error pattern is then used as the behavior regularization to penalize the final loss, as shown from line 10 to line 13.

## B  Implementation Details

In this section, we provide detailed training methods for relational reasoning tasks and reinforcement learning tasks. The hyper-parameters of the network structure and training settings are also provided. We conduct our experiments on a Ubuntu 18.05 server with 48 cores of Intel Xeon Siver 4214 CPU, 4 NVIDIA Quadro RTX 8000 GPUs, 2 NVIDIA Quadro RTX 6000 GPUs, and 252GB RAM.

### B.1  Training Settings

**Training Method.**  Both NLM (Dong et al., 2018) and NLM w/ FRGR are trained using Adam optimizer (Kingma & Ba, 2014) with a 0.005 learning rate. For all the relational reasoning tasks, the Softmax Cross Entropy is used as the loss function. For reinforcement learning tasks, the REINFORCE (Williams, 1992) algorithm is used for optimization. Similar to NLM, the policy entropy term is also added to the loss function. By adding the behavioral regularization term, parameters $\theta$ of the RL policy $\pi$ is updated via:

$$\theta' = \theta + \eta[\gamma_r^t \nabla_\theta \log \pi_\theta Q_{\pi_\theta}(s_t, a_t) + \beta \nabla_\theta H(\pi_\theta) - \gamma \nabla \mathcal{L}_{\text{BEH}}], \tag{1}$$

where $\eta$ is the learning rate, $\gamma_r^t$ is the discounted reward at time step $t$, $H$ is the entropy regularization, $\beta$ is the discount factor to control the entropy, $s_t$ and $a_t$ are the state and action at time step $t$. Across all the environments, a positive of $+1.00$ will be given to the agent. To encourage the agent to use as few moves as possible, a negative reward of $-0.01$ is given for each action taken.

---

**Algorithm 1:** FRGR regularization framework for reinforcement learning tasks.

**Input:** Training number of episodes $Epi$, Neural induction model $\pi$, Erroneous behavior list $E$, Erroneous behavior list size $\tau$, Error pattern $\epsilon$, Behavior extractor $\rho$, Regulatory coefficient $\gamma$, Learning rate $\eta$, Model weights $\theta$, REINFORCE loss $\mathcal{L}_{\text{REI}}$, Behavioral loss $\mathcal{L}_{\text{BEH}}$

1   Initialization: $c \leftarrow 0, i \leftarrow 0, e \leftarrow 0, E \leftarrow MaxHeap(\tau)$
    /* Error Pattern Mining */
2   **while** $e < Epi$ **do**
3      Running Policy $\pi_\theta$ for $T$ steps
4      Collecting trajectory $Traj = \{(s_t, a_t, r_t, \omega_t)\}_{0...T}$
5      Calculating the discounted return $\{G_1, G_2, \ldots, G_T\}$
6      **if** $Traj\ Fails$ **then**
7        Update $E$ according to $\{(G_t, \omega_t)\}_{0...T}$
8      **end**
9      $\epsilon \leftarrow Apriori(E)$
      /* Update with behavioral regularization */
10     $m \leftarrow \omega \cap \epsilon$                   // error pattern matching
11     $\mathcal{L}_{\text{BEH}} \leftarrow \sum_{\nu \in m} |\theta_\nu|$
12     $\mathcal{L} \leftarrow \mathcal{L}_{REI} - \gamma \cdot \mathcal{L}_{\text{BEH}}$
13     $\theta \leftarrow \theta + \eta \nabla_\Theta \mathcal{L}$
14     $e \leftarrow e + 1$
15   **end**

---

**Curriculum Learning.** For reinforcement learning tasks, curriculum learning (Bengio et al., 2009; Dong et al., 2018) is also applied. The training instances are grouped into lessons according to their complexity. The number of objects in the environment is considered an indicator of complexity. The model will start with a simple lesson and gradually increase the difficulty when the model passes the exam. The exam will be taken when the model is well-trained on the current lesson, i.e., the accuracy reaches a certain threshold. Specifically, during the lesson, all failed and successful environments will be recorded. The training examples will be sampled from the successful environments with the probability of $\Omega$ and failed environments with the probability of $1 - \Omega$.

### B.2   HYPERPARAMETERS

**Hyperparameters for relational reasoning tasks.** The details of the network structure of both NLM and NLM w/ FRGR are shown in Table 1. For each computation unit, no hidden layer is used and the number of intermediate predicates (hidden dimension) is set to be 8 for all the benchmarks. Specifically, the residual means the input predicates are concatenated to the output predicates of each computation unit. For the data-rich scenario, the examples are divided into 500 epochs, each containing different samples. For the data-scarce scenario, the examples are the same for each epoch. The batch size is set to be 4 across all the experiments. The regulatory coefficient $\gamma$ is set to be 0.99 and erroneous behavior list size $\tau$ is set to be 100 for all tasks.

**Hyperparameters for reinforcement learning tasks.** Table 2 shows the details of the network structure and hyperparameters for reinforcement learning tasks. Each training batch contains one episode. Similarly, no hidden layer is used and the number of intermediate predicates is also set to be 8. Residual linkage is applied for all the RL tasks. Specifically, for curriculum learning, the induction model starts from a small number of objects and gradually advances to a larger number. For example, the first lesson for the Sorting task contains environments with 2 objects. The second lesson contains environments with 3 objects, and the final lesson contains environments with 10 objects. For the data-rich scenario, the environments are different for each lesson taken. For the data-scarce setting, the environments are the same for the same level of lessons. The regulatory coefficient $\gamma$ is set to be 0.99 and erroneous behavior list size $\tau$ is set to be 100 for all tasks.

Table 1: The details of the network structure for the NLM and the NLM w/ FRGR models for the relational reasoning tasks. Residual indicates the use of Input/Output residual links.

|  | Tasks | Depth | Breadth | Residual | Examples (Data-rich) | Examples (Data-scarce) |
|---|---|---|---|---|---|---|
| | HasFather | 4 | 3 | No | 50,000 | 100 |
| | HasSister | 4 | 3 | No | 50,000 | 100 |
| Family Tree | IsGrandparent | 4 | 3 | No | 100,000 | 200 |
| | IsUncle | 4 | 3 | No | 100,000 | 200 |
| | IsMGUncle | 4 | 3 | No | 200,000 | 400 |
| | 1-Outdegree | 4 | 3 | No | 50,000 | 100 |
| General Graph | 2-Outdegree | 5 | 4 | Yes | 100,000 | 200 |
| | 4-Connectivity | 4 | 3 | No | 50,000 | 100 |
| | 6-Connectivity | 8 | 3 | Yes | 50,000 | 100 |

Table 2: The details of the network structure and hyperparameters for the NLM and the NLM w/ FRGR models for the reinforcement learning tasks. The Lessons indicate the different levels of lessons used for training.

| Tasks | Depth | Breadth | Residual | Lessons | $\Omega$ | Epochs | Total Episodes (Data-rich) | Total Episodes (Data-scarce) |
|---|---|---|---|---|---|---|---|---|
| Sorting | 3 | 2 | Yes | [4,10] | 0.5 | 50 | 1,000 | 140 |
| Path | 5 | 3 | Yes | [3,12] | 0.5 | 400 | 24,000 | 600 |
| BlocksWorld | 7 | 2 | Yes | [2,12] | 0.6 | 500 | 50,000 | 1,100 |

## C  DETAILED DESCRIPTIONS OF BENCHMARKS

**Relational reasoning tasks.**  For the Family Tree Reasoning tasks, a family tree consisting of $n$ family members is given to the neural program induction model. The relationships between each pair of family members are represented by the following predicates: `IsFather`, `IsMother`, `IsSon`, and `IsDaughter`. The goal of this task is to learn the properties of family members or the relations between them: `HasFather`, `HasSister`, `IsGranparent`, `IsUncle`, `IsMGUncle` (defined as maternal great uncle).

For the General Graph Reasoning tasks, an undirected graph comprising $n$ nodes is given to the induction model. The relationships between nodes are represented by the predicate `HasEdge`. The goal of this task is to acquire the properties of the nodes or the relations between them: $k$-`Connectivity` and $k$-`OutDegree`. Specifically for $k$-`Connectivity`, the induction model is expected to determine whether two nodes can be connected by a path with at most $k$ edges; for $k$-`OutDegree`, it is expected to classify whether the out-degree of the node equals to $k$.

**Reinforcement learning tasks.**  For the Sorting task, an array of length $m$ is given to the induction model, and it needs to learn the `swap` predicate (i.e., swapping two integers in the array) to sort the list in ascending order. The index relation predicates (`SmallerIndex`, `SameIndex`, `LargerIndex`) and numerical relations predicates (`SmallerNumber`, `SameNumber`, `LargerNumber`) are grounded with each pair of integers and are used as premises for the induction model. For the Path environment, given an undirected graph, the induction model needs to find a path between the start node $s$ and the end node $e$, which are represented by two unary predicates (`IsStart`, `IsEnd`). The path is learned by choosing the next node to go to (i.e., learn the `Next` predicate) until reaching the end node $e$. The Blocks World task includes two worlds: an initial world and a target world, both of which contain $m$ objects ($m-1$ cubes and 1 ground). The induction model has to learn how to move (the binary predicate `Move`) the objects to change the world from the initial setting to the target setting. Each object is represented by four characteristics: `world_id`, `cube_id`,

`coordinate_x`, and `coordinate_y`. The binary relations (represented by binary predicates) of all the above four characteristics, listed in the following, are given to the induction model as input: `SmallerWorldID`, `SameWorldID`, `LargerWorldID`, `SmallerCubeID`, `SameCubeID`, `LargerCubeID`, `SmallerX`, `SameX`, `LargerX`, `SmallerY`, `SameY`, `LargerY`.

## D    DETAILED DESCRIPTIONS OF NEURAL LOGIC MACHINE

**Expansion**    The Expansion operation constructs a new predicate $q$ from $p$ by introducing a new variable $x_{b+1}$, which provides more flexibility for NLM to build expressive formulas:

$$\forall x_{b+1}\ q\left(x_1, x_2, \cdots, x_b, x_{b+1}\right) \leftarrow p\left(x_1, x_2, \cdots, x_b\right).$$

As an example, a new predicate $p_3$ can be constructed via the Expansion operation and Boolean logic operation, as shown below.

$$
\begin{aligned}
&\forall z q_1\left(x, z\right) \leftarrow p_1\left(x\right) && \text{(Expansion)} \\
&\forall z q_2\left(y, z\right) \leftarrow p_2\left(y\right) && \text{(Expansion)} \\
&p_3\left(x, y\right) \leftarrow q_1\left(x, z\right) \wedge q_2\left(z, y\right) && \text{(Boolean Logic)}
\end{aligned}
$$

More specifically, considering the set of $N^b$ $b$-ary predicates are represented by the tensor of shape $[m^{\underline{b}}, N^b]$, the Expansion is done by stacking the tensor $(m - b)$ times. This results in a new tensor of the shape $[m^{\underline{b+1}}, N^b]$.

**Reduction**    The Reduction operation constructs a new predicate $q$ from $p$ by reducing a variable in $p$ via the logic quantifier. The logic quantifier consists of two meta-rules:

$$
\begin{aligned}
q\left(x_1, x_2, \cdots, x_b\right) &\leftarrow \forall x_{b+1}\ p\left(x_1, x_2, \cdots, x_b, x_{b+1}\right) \\
q\left(x_1, x_2, \cdots, x_b\right) &\leftarrow \exists x_{b+1}\ p\left(x_1, x_2, \cdots, x_b, x_{b+1}\right)
\end{aligned}
$$

As an example, a new predicate $p_2$ can be constructed via the Reduction operation and Boolean logic operation, as shown below:

$$
\begin{aligned}
&q_1\left(x\right) \leftarrow \forall z p_1\left(x, z\right) && \text{(Reduction)} \\
&p_2\left(x\right) \leftarrow q_1\left(x\right) \wedge p_3\left(x\right) && \text{(Boolean Logic)}
\end{aligned}
$$

More specifically, considering the set of $N^{b+1}$ $b+1$-ary predicates are represented by the tensor of shape $[m^{\underline{b+1}}, N^{b+1}]$, the Reduction operation takes the maximum (for $\exists$) or minimum (for $\forall$) value along the $x_{b+1}$ dimension and then stack these two tensors. This results in a new tensor of the shape $[m^{\underline{b}}, 2N^{b+1}]$.

**Permutation**    Given an $b-$ary predicate $p(x_0, x_1, \cdots, x_b)$, the Permutation operation generates $b!$ grounding values on the same set of objects. For example, consider a ternary predicate $p(x, y, z)$ grounded on three objects $a, b, c$. The Permutation operation generates the grounding values of all the permutations of the variables: $p(a, b, c), p(a, c, b), p(b, a, c), p(b, c, a), p(c, a, b), p(c, b, a)$. Therefore, considering the set of $N^b$ $b-$ary predicates represented by the tensor of shape $[m^{\underline{b}}, N^b]$, the Permutation operation generates $b!$ such tensors, resulting in a tensor of shape $[m^{\underline{b}}, b! \times N^b]$.

**Logic Computation**    For each computation at depth (layer) $d$, breadth (arity) $b$, the input tensor $In_d^b$ comes from the output from last depth $d - 1$ and breadth $b - 1, b, b + 1(Out_{d-1}^{b-1}, Out_{d-1}^b, Out_{d-1}^{b+1})$:

$$In_d^b = \text{Concat}(\text{Expansion}(Out_{d-1}^{b-1}), Out_{d-1}^b, \text{Reducion}(Out_{d-1}^{b+1})).$$

The neural boolean logic is computed via a multi-layer perceptron to derive newly invented predicates $Out_d^b$:

$$Out_d^b = \sigma(\text{MLP}(\text{Permutation}(In_d^b))),$$

where $\sigma(\cdot)$ is the nonlinearity function.