# OpenReview forum: "Improving Neural Program Induction by Reflecting on Failures"
_ICLR.cc/2024/Conference — Submitted to ICLR 2024_

### Official Review · Reviewer_bWPz · 2023-10-25

**Soundness:** 2 fair
**Presentation:** 2 fair
**Contribution:** 3 good
**Rating:** 3
**Confidence:** 3

**Summary:**

The paper proposes FRGR, an approach to improve training of neural logic models (NLMs). The intuition behind FRGR is to guide the NLM away from incorrect solutions found in prior training iterations. FRGR works by adding an additional regularization term that penalizes the magnitude of weights that in prior training iterations were high-magnitude connections between layers of the NLM. The paper evaluates FRGR on a range of tasks, finding that it generally results in faster training and better generalization than standard NLM training without the FRGR regularization term.

**Strengths:**

* The paper addresses an interesting domain and problem of improving training (both accuracy and efficiency) of neural logic programs
* The core of the paper is an interesting insight of adding a term to avoid past errors observed when training the NLM
* The evaluation results (taken at face value, though see weaknesses below) are quite strong.

**Weaknesses:**

* I find Section 2.2 very hard to understand. I have some background in logic programming, but none in neural logic machines. Specifically, I was unsure on the following points:
  * "First, given a set of objects $\mathcal{U}$ of size $m$..." -- it is not clear what an object is, or what it means for an object to have a size.
  * "the NLM model first grounds $p$ on $\mathcal{U}$, which derives a tensor representation of $p^\mathcal{U}$..." -- I'm not sure what it means to ground $p$ in this context.
  * I broadly found the discussion of Figure 2 to be hard to understand
* I also find Section 3 hard to understand:
  * Algorithm 1, line 4/5: what is the difference between $f$ and $f_\theta$?
  * "$b$ and $d$ denote the number of depth and breadth of the NLM model architecture respectively" -- is this backwards, or does $b$ refer to depth and $d$ refer to breadth?
* Broadly, I am not convinced by the high level intuition behind the approach. My understanding of the approach is to penalize the magnitude of weights that in prior iterations were high-magnitude connections between layers of the NLM. However, I don't see why this should necessarily lead to a better solution than standard gradient descent: if the set of predicates learned by the NLM is the same as in past iterations, then standard gradient descent should penalize incorrect large-magnitude weights; if the set of predicates is different than in past iterations, then this is penalizing using predicates that may not be incorrect. Essentially, this seems to have the effect of either (1) increasing the effective learning rate, and (2) adding weight decay. I do want to note that a sufficiently strong evaluation would convince me to raise my score despite not being convinced by the high level intuition. However, the evaluation does not pass that bar (see below).
* Evaluation:
  * Hyperparameters: as mentioned above, one of the effects of the additional term in the loss function is increasing the effective learning rate. The paper does not provide evidence that the proposed approach outperforms a baseline with a well tuned learning rate. The paper also does not justify (or state) other hyperparameter choices.
    * The paper does not state how hyperparameters are selected. In particular, hyperparameters are (as best I can tell) identical between the training methods, which may be an unfair comparison.
    * I also cannot find the value of the hyperparameter $\gamma$ coefficient for the regularization, the value of the history size $\tau$, or how these values were tuned.
  * Regularization: as mentioned above, another effect of the additional term in the loss function is weight decay. The paper does not provide evidence that the proposed approach outperforms a baseline with weight decay (or other regularization).

**Questions:**

* How are hyperparameters for the evaluation tuned? What are $\gamma$ and $\tau$ set to?
* Does NLM w/ FRGR still outperform standard NLM when both approaches are given the same budget for hyperparameter tuning? When NLM is ran with weight decay?

---

> ### Author Response · Authors · 2023-11-16
> **Response to Reviewer bWPz [1/3]**
>
> We thank the reviewer for the detailed review and comments. Please see the following for our response!
>
> ### 1. I find Section 2.2 very hard to understand. I have some background in logic programming, but none in neural logic machines. "First, given a set of objects U f size m “ it is not clear what an object is, or what it means for an object to have a size "the NLM model first grounds p on U, which derives a tensor representation of p_U..." -- I'm not sure what it means to ground p in this context.
>
> Object refers to an entity or element within a domain of discourse, which is a fundamental and broadly accepted concept of first-order logic [1]. “a set of objects $U$ of size $m$ ” means there is a set $U$, and $U$ contains m objects: $|U|=m$. Grounding is a frequently used concept in first-order logic which refers to the process of creating a variable-free first-order formula equivalent to a first-order sentence [2][3]. Thus, “ grounds p on $U$” denotes the process of instantiating the variables ($\alpha_1, \dots, \alpha_n$) of atom p($\alpha_1, \dots, \alpha_n$) with the corresponding truth values of the attributes/relations of the object set $U$ (of size $m$) which results in m truth values for the m respective objects. For instance, for the predicate IsFather( $X$, $Y$), and a family of three family members: $U={A, B, C}$. Grounding IsFather on $U$ results in a set of truth values of the ground atoms: IsFather($A$, $B$), IsFather($A$, $C$), IsFather($B$, $C$).
>
> ### 2. Algorithm 1, line 4/5: what is the difference between $f$ and $f_\theta$ ?" $b$ and $d$ denote the number of depth and breadth of the NLM model architecture respectively" -- is this backwards, or does refer to depth and refer to breadth?
>
> $f$ and $f_\theta$ denotes the same neural model, \theta denotes the the model $f$ is parameterized with the parameters set $\theta$. $b$ denotes that model breadth and $d$ denotes the model depth. We will update these in our paper.
>
> ### 3. However, I don't see why this should necessarily lead to a better solution than standard gradient descent: if the set of predicates learned by the NLM is the same as in past iterations, then standard gradient descent should penalize incorrect large-magnitude weights; if the set of predicates is different than in past iterations, then this is penalizing using predicates that may not be incorrect.
>
> Our approach is inspired by the success of provenance information for SAT-based program synthesis, in which previous error information is leveraged to reduce future search space. We empirically demonstrate that the vanilla training loss is not effective enough to capture model's erroneous behavior pattern and thus results in a ungeneralizable solution as well as low training efficiency. We further conduct a post hoc interpretation analysis to demonstrate the effectiveness of FRGR in refraining model from repeating erroneous behavior. We present an error pattern which FRGR manages to mitigate. Concretely, for the IsMGUncle and IsUncle tasks, their ground-truth programs only involve using binary predicates. The ground-truth programs are shown as follows:
>
> IsUncle:
>
> $$
>  \operatorname{IsUncle}(X, Y) \leftarrow \exists Z,  ((\operatorname{IsMother}(X, Z) \wedge \operatorname{IsBrother}(Z, Y))) \\
>  \vee(\operatorname{IsFather}(X, Z) \wedge \operatorname{IsBrother}(Z, Y))
>  $$
>
>  $$
> \operatorname{IsBrother}(X, Y) \leftarrow \exists Z,  ((\operatorname{IsSon}(Z, Y) \wedge \operatorname{IsSon}(Z, X)) \\
>  \vee(\operatorname{IsSon}(Z, Y) \wedge \operatorname{IsDaughter}(Z, X)))
> $$
>
> IsMGUncle:
> $$
> \operatorname{IsMGUncle}(X, Y) \leftarrow \exists Z,(\operatorname{IsMother}(X, Z) \wedge \operatorname{Is} \operatorname{Uncle}(Z, Y))
> $$
>
> Therefore, it is considered a erroneous behavior if the model strongly attributes its induction based on unary predicates. We calculate the ratio of all output binary predicates whose most attributed input predicate is unary during the training process. We evenly sample 5 epochs from the training process till nlm w/ FRGR converge to a optimal solution. The results are shown in Table 1. It is obvious that with the use of FRGR, the ratio of unary predicates are rapidly decreased compared with the vanilla NLM. The results indicate the FRGR is effectively in recognizing repetitious error pattern during the training process and actively regularizing the model based on it, which results in higher training efficiency and better generalization.
>
> ||**IsMGUncle**|||||
> |:--|:-|:--|:--|:--|:--|
> |NLM|0.417|0.444|0.426|0.486|0.394|
> |NLM w/ FRGR|0.417|0.426|0.454|0.394|0.356|
> ||**IsUncle**|||||
> |NLM|0.433|0.458|0.450|0.433|0.483|
> |NLM w/ FRGR|0.433|0.442|0.467|0.358|0.375|
>
> [**Table 1**: interpretation analysis.]
>
> This process is similar to that of the provenance information used in the SAT-based synthesis, in which the solver's error is actively summarized based on previous derivation trees and is used to reduce future search space.

---

> ### Author Response · Authors · 2023-11-16
> **Response to Reviewer bWPz [2/3]**
>
> ### 4. Hyperparameters: as mentioned above, one of the effects of the additional term in the loss function is increasing the effective learning rate. The paper does not provide evidence that the proposed approach outperforms a baseline with a well tuned learning rate. The paper also does not justify (or state) other hyperparameter choices. The paper does not state how hyperparameters are selected. In particular, hyperparameters are (as best I can tell) identical between the training methods, which may be an unfair comparison. I also cannot find the value of the hyperparameter coefficient for the regularization, the value of the history size or how these values were tuned.
>
> The details of the training settings and hyperparameter choices are delivered in Appendix B&C with clear reference on page 6&7 of the paper. We state that all the training settings and hyperparameter choices for the NLM model strictly follow previous work [3] instead of being cherry-picked by ourselves.
> We have to state that it is inappropriate for the reviewer to state that: “hyperparameters are (as best I can tell) identical between the training methods, which may be an unfair comparison.” as it is an extremely common practice to set hyperparameters (such as learning rate) the same for the evaluated baseline models. Please feel free to refer to the cited literature for details [4][5]. Nevertheless, we conduct the hyperparameter analysis of the learning rate, the results are shown in Table 2: We conduct hyperparameter analysis for the IsGrandparent task from the family tree benchmark and 2-outdegree from the graph reasoning benchmark. Specifically, we experiment with the following learning rates: 0.005 (default of NLM), 0.01, 0.05, 0.1 for both NLM and NLM w/ FRGR.
> For IsGrandparent, the results are shown in the table below. Concretely, for the original NLM model, it performs the best under the default setting (0.005 learning rate) while for NLM w/ FRGR, it performs even better under the 0.1 learning rate. Specifically, the learning iterations can be decreased by +49.82% (from 55.80 to 28.00).
>
> | IsGrandparent|lr |grad-rate|n=20 |n=100|Iterations  |
> |:--|:--|:--|:-|:-|:--|
> |NLM|0.005|100.00% |100.00%|100.00%|96.20±21.87|
> |NLM w/ FRGR|0.1|100.00% |100.00%|100.00%|28.00±6.73 |
>
> [**Table 2**: learning rate analysis.]
>
> For 2-Outdegree, the best results for both the NLM and NLM w/ FRGR are achieved under the default learning rate (0.005), which are the same results shown in Table 1 of our paper. Therefore, with the same budget of hyperparameter tuning, NLM w/ FRGR steadily outperforms the original NLM.

---

> ### Author Response · Authors · 2023-11-16
> **Response to Reviewer bWPz [3/3]**
>
> ### 5. Regularization: as mentioned above, another effect of the additional term in the loss function is weight decay. The paper does not provide evidence that the proposed approach outperforms a baseline with weight decay (or other regularization).
>
> The weight decay mechanism is already implemented in the original NLM model, which further demonstrates the effectiveness of FRGR’s effectiveness over this regularization approach. Besides, we further conduct an ablation study by comparing our approach with a random regularization baseline (i.e., randomly penalizing the same percentage of the model’s weights as that of FRGR), denoted as NLM w/ random. The results are shown in Table 3:
>
> |2-Outdegree|grad-rate|n=10|n=20|Iterations|
> |:-|:--|:-|:--|:--|
> |NLM|90.00%|96.52%±0.10|90.80%±0.28|77.9±150.5|
> |NLM w/ random|80.00%|82.10%±0.36|81.18%±0.38|114±203.70|
> |NLM w/ FRGR|100.00%|100.00%±0.00|100.00%±0.00|13.40±11.13|
>
> [**Table 3**: regularization comparisons.]
>
> The results illustrate that by adding random regularization, the model's performance decreases compared to that of the original NLM. Specifically, compared to the original NLM model, the grad-ratio decreases by +10.00%, and the performance decreases by -14.42%.
> While with FRGR, it is effective in improving grad-ratio, performance, generalization, and training efficiency.
> The results demonstrate that the vanilla training loss, random regularization as well as weight decay regularization are not sufficient enough to capture the repetitious error pattern during training thus incapable of improving the NLM model.
>
> ### 6. How are hyperparameters for the evaluation tuned? What are $\gamma$ and $\tau$ set to?
>
> We haven’t conducted specific hyperparameter tuning for our method. And we directly adopt the hyperparameter settings from the original NLM paper and used them for all evaluated methods. $\gamma$ is set to be 0.99 and $\tau$ is set to be 100 for all tasks.
>
> [1] https://www.cs.rice.edu/~vardi/comp409/lec14.pdf
>
> [2] Migimatsu, Toki, and Jeannette Bohg. "Grounding predicates through actions." 2022 International Conference on Robotics and Automation (ICRA). IEEE, 2022.
>
> [3] Dong, Honghua, et al. "Neural Logic Machines." International Conference on Learning Representations. 2018.
>
> [4] Kim, Donggyun, et al. "Universal Few-shot Learning of Dense Prediction Tasks with Visual Token Matching." The Eleventh International Conference on Learning Representations. 2022. (Outstanding Paper)
>
> [5] Riad, Rachid, et al. "Learning Strides in Convolutional Neural Networks." International Conference on Learning Representations. 2021. (Outstanding Paper)

---

> ### Author Response · Authors · 2023-11-20
>
> We would appreciate it if the reviewer can be more involved in the discussion (Nov 10-22) and let us know whether the response has clarified the corresponding questions.

---

> > ### Comment · Reviewer_JknR · 2023-11-20
> >
> > I had similar concerns as this reviewer to be confused why the augmented feedback of penalizing weights used in erroneous solutions is better than SGD. My intuition is that SGD updates may not "generalize" as well as the penalized weights, which use a crude type of symbolic reasoning to determine which paths should be penalized. However, I'm not very confident in this intuition

---

> > > ### Author Response · Authors · 2023-11-20
> > >
> > > We agree with the reviewer’s point. Specifically, we think this is because the vanilla training loss is not effective and efficient enough to capture the model's erroneous behavior pattern. As a result, the model would make mistakes of similar patterns in future iterations, and thus make it more likely to converge to a local optimum that is ungeneralizable (\ie~a solution that is either redundant or erroneous as it would rely on undesired predicates). And by applying pattern mining techniques to the model’s previous erroneous behaviors, our method is able to identify the error pattern more efficiently and therefore manages to refrain the model from repeating mistakes of similar patterns in the future training process. This enables the model to converge to a generalizable solution more quickly and with greater likelihood.
> > >
> > > This explanation can be reflected by our interpretation analysis on the IsUncle and IsMGUncle tasks (whose ground-truth programs only involve using binary predicates) as shown above. As the MLP functions as the implementation of the Horn clauses [1]: $\forall x \hat{p}(x) \leftarrow p_1(x) \wedge \dots \wedge p_n(x)$. The most activated weights of an output neuron (output predicate $\hat{p}(x)$) represent the most attributed input predicates within the conjunction of the rule body.
> > > If the most attributed predicates are frequently unary predicates, the solution would either be redundant or erroneous. We calculate the ratio of the most attributed input predicates that are unary. It is obvious from the results in Table 1 (shown above) that for the original NLM model (vanilla training loss only), the ratio is high and steady throughout the training process. Whereas when regularized with our approach (NLM w/ FRGR), the ratio rapidly decreases and the model manages to converge to a generalizable solution much faster.
> > >
> > > [1] Dong, Honghua, et al. "Neural Logic Machines." International Conference on Learning Representations. 2018.

---

> ### Author Response · Authors · 2023-11-21
>
> We would appreciate it if the reviewer could let us know whether the response has clarified the corresponding questions as it is approaching the Author/Reviewer Discussion period deadline (Nov 22) and it has already been 5 days since we posted our response (Nov 16).

---

> > ### Comment · Reviewer_bWPz · 2023-11-21
> >
> > Thanks to the authors for the detailed rebuttal. Based on this rebuttal, I'm currently planning to keep my score as-is. I do think the paper is interesting and the results are strong (assuming they hold up to more rigorous scrutiny, which is not currently possible with the description of the technique and its evaluation in the paper). But though the authors have clarified the approach and presented more evidence of its effectiveness, I still believe that the paper needs another round of revision (and fresh reviews) before acceptance.
> >
> > My main critiques of the paper are of the clarity of the writing, and of the strength of evidence for the presented approach.
> >
> > **Clarity:** This was a common issue among all reviewers, and while I appreciate the clarifications offered in the response and the promises to update the paper, I don't feel comfortable raising my score on these grounds without a revised version of the paper that is significantly more clear.
> >
> > **Strength of evidence:** Given the lack of theoretical results and that other reviewers also do not have an intuition for why the proposed approach should work, this paper requires strong empirical evidence to support the claim "that FRGR can considerably improve the training efficiency… of the neural program induction model". However even with the additional experiments and details provided on hyperparameters in the rebuttal, I still believe the paper needs another revision and set of reviews to provide sufficiently solid evidence on this point. In particular, the paper does not provide enough details on hyperparameters and their tuning to reproduce the experimental results or to give confidence that a practitioner or follow-up paper would be able to apply this technique with the same success as in this paper (relative to the baseline).
> >
> > The authors state that it is inappropriate for me to ask about hyperparameter tuning for the baseline. I strongly disagree with this. The original submission did not provide any evidence that the chosen hyperparameter settings were reasonable for the baseline (for the settings not explicitly evaluated in Dong et al., 2018) or discuss the resources required to choose the hyperparameters for FRGR (e.g., the response to Reviewer 7KpG shows that the choice of tau=100 leads to significant increases in performance on 6-connectivity relative to tau=75 or 125; the paper and response do not discuss how this choice was initially made to set tau to 100, rather than e.g. 1, 10, or 1000, or how gamma was set to 0.99). The rebuttal provides learning rate experiments, but still does not provide a complete description of how all hyperparameters were tuned.
> >
> > Further, many hyperparameters and methodological details are still left unclear. For instance, the rebuttal states that  "the weight decay mechanism is already implemented in the original NLM model". The paper does not mention this, say what the weight decay hyperparameter is, or how it was tuned. Especially given that the paper is comparing an alternative regularization term, this is a crucial detail, and leaving it out significantly decreases my confidence in the results.
> >
> > In writing this response, I also noticed that the results for vanilla NLM reported in the paper (Table 1) are significantly worse than the results for vanilla NLM reported in Dong et al., 2018: specifically, Dong et al report 100% accuracy for all family tree and graph reasoning tasks. Maybe I'm missing something in the paper, but why are the NLM results in this paper much worse than the NLM results in Dong et al, 2018?

---

> > > ### Author Response · Authors · 2023-11-21
> > > **Response to Reviewer bWPz**
> > >
> > > Thanks for the response and follow-up discussion.
> > >
> > > ## Clarity:
> > > We have uploaded the revised version with updated parts in Section 2 and 3 to explain NLM model design more plainly in the preliminary and we use a concrete case to explain the rationale of FRGR in the methodology section. Besides, we have revised all the mentioned details by all the reviewers as well. The revised parts are highlighted in blue. We hope the reviewer to inspect our revised version and take it into consideration for the rating.
> > >
> > > ## Evidence
> > > Regarding reproducibility, we only introduced two new hyperparameters, $\tau$, and $\gamma$, which have been clearly stated in Appendix B.2. We have also released our code base, as clearly stated in the paper. Regarding the hyperparameters FRGR introduced ($\tau$ and $\gamma$), we haven’t conducted any tuning of these two hyperparameters and it’s just a choice of instinct.
> > > Regarding all other hyperparameters like learning rate, network structure, batch size, etc., we strictly follow the settings of NLM, which are clearly stated in their Appendix and code base. And we strictly followed the NLM’s appendix and their code base. **We do not see a reason why we need to provide evidence that the chosen hyperparameter settings are reasonable for the baseline as we strictly follow the baseline’s original setup. If we change the hyperparameters of the baseline, we believe it will draw more questions from the reviewers saying that why not follow the settings of the baseline method.**
> > >
> > > Regarding the weight decay, we use the built-in weight decay of AdamW optimizer, which is the default optimizer in the code base of NLM, and the weight decay ratio is set to the default value 0.01. We seriously state that we have not changed anything. Please feel free to check our code base and NLM’s code base. Again, how the hyperparameters were tuned or chosen was not explained or illustrated in the NLM paper. Therefore, we do not see why this is crucial and important to explain all the details of the hyperparameters.
> > >
> > > ## Results different from NLM
> > > The original NLM paper reported the results of the best model obtained over 10 random seeds, while we reported the average results over 10 random seeds instead of selecting the best. We deem deemed this a better way to evaluate the models' performance. To further clarify this, as illustrated in Appendix B.2 (Table 4) of the original NLM paper, the NLM model only achieves a graduation ratio of 20%, which indicates that there are only 2 out of 10 runs that achieve 100% accuracy on the training instances. Thus, averaging the results from all these 10 runs leads to an accuracy that is less than 100%.

---

> > > ### Author Response · Authors · 2023-11-22
> > > **New Response to Reviewer bWPz**
> > >
> > > We got the impression that the reviewer was probably worrying that we deliberately tuned the hyperparameters to decrease the performance of NLM to support our claim.
> > >
> > > **We have to emphasize that we did not conduct any hyperparameter tuning to jeopardize to performance of NLM.**
> > > More specifically, our FRGR only introduces two new hyperparameters $\tau$ and $\mu$, which are not used by the original NLM model. Namely, NLM does not have $\tau$ and $\mu$ as hyperparameters, therefore these two hyperparameters have no effect on NLM.
> > >
> > > Regarding all other hyperparameters, like breadth, depth, learning rate, optimizer, weight decay, batch size, epoch size, etc., **we strictly follow the settings of the original NLM paper to reproduce their results and get our results, as stated in their appendix and code base and ours.** Please feel free to check the Appendix and code base of both our method and the NLM paper for the hyperparameters. We do not discuss these hyperparameters in detail as they are not the key point of our paper.
> > >
> > > Regarding the different results from the original NLM paper. The original NLM paper reported the results of the best model obtained over 10 random seeds, while we reported the average results over 10 random seeds instead of selecting the best. We deem deemed this a better way to evaluate the models' performance. To further clarify this, as illustrated in Appendix B.2 (Table 4) of the original NLM paper, the NLM model only achieves a graduation ratio of 20% for the IsMGUncle task, which indicates that there are only 2 out of 10 runs that achieve 100% accuracy on the training instances. Thus, averaging the results from all these 10 runs leads to an accuracy that is less than 100%.
> > >
> > > Regarding the tuning of $\tau$ and $\mu$, we did not conduct any tuning initially. For $\tau$, we have conducted an ablation study in the rebuttal session. The results show that the selection of $\tau$ has some effects on the performance of FRGR, however, all the results still outperform the results of NLM. For $\mu$, we just set it to 0.99 out of distinct. However, as $\mu$ goes lower, the model will gradually downgrade to the original NLM model (when $\mu$ equals to zero). So the lower bound of tuning $\mu$ is the original NLM model.

---

### Official Review · Reviewer_YeEy · 2023-10-30

**Soundness:** 3 good
**Presentation:** 1 poor
**Contribution:** 3 good
**Rating:** 5
**Confidence:** 3

**Summary:**

In this paper, the authors introduce a regularisation scheme for Neural Logic Machines, aimed at improving their performance on Neural Program Induction tasks. The scheme works by "recording" the errors committed during training and regularising the neural networks weights associated with them via $L_1$-type loss. The authors report improved performance (w.r.t a NLM trained without using their scheme) and data efficiency over a range of program induction tasks.

**Strengths:**

- The proposed regularisation scheme appears logically sound and ultimately relatively simple to implement.
- The authors demonstrate that it does improve performance and data efficiency on a range of tasks.

**Weaknesses:**

- The paper is written for a specialist audience, with little context or background given about NLP, about what the challenges in it are, about how ML methods can help solve it, or even just about what the tasks look like. In general, a formulation of the problem as a task that can be used to train a neural network on a certain experience and with a certain performance metric is omitted completely and taken for granted. As a result, most readers (as well as this referee) will not be able to get much from this paper.
- Because of this, it's hard (though not impossible, see below) for me to assess the scientific content of this paper and the scope of its contribution to the field. Bit I nevertheless feel that the shortcomings on the presentation side are still sufficient to recommend that it be rejected or at least substantially revised.
- The authors' method appears to work as intended, but also to be quite narrow in applicability: it only presumably works on NLMs (since it requires direct control of the weights) and only for the particular task considered by the authors. At first sight, the scope of the contribution appears very limited.
- The authors refer to their scheme as a constraint on the weights, which seems to imply thet an hard constraint is imposed on them, whilst the scheme is more properly defined as a regularisation strategy, or a "soft" constraint.

**Questions:**

My main recommendation would be for the authors to provide context on NLP, NLMs, and on how their task can be solved via ML methods to begin with, and on how one would train a neural net to do so. Without such context, it's for me impossible to do justice to the content of this paper, but it is also going to be impossible for a mainstream ML reader (such as most of those attending ICLR) to profit from the paper to begin with.

On the contribution side, the authors should prove that their regularisation scheme does have any general applicability beyond NLP tasks carried out with NLMs.

Unless both of these shortcomings can be suitably addressed, the implication will be that this work is more suited for publication in a specialised venue rather than a broad conference such as ICLR.

## Post-rebuttal edit:

The authors' rebuttal did convince me that my assertion of the general applicability of their method was erroneous and that "NLP tasks" was a somewhat misleading designation on my part. The architecture that the paper's regularisation method is applied to (the Neural Logic Machine or NLM) can indeed be used for multiple reasoning benchmarks of interest, which the authors do in their submission. In reason of this, I have revised my contribution score to "good".

I think that the papers still suffers from unclear writing, which results in the context of the author's work not being properly outlined for anyone not already familiar with it. The authors have not seemingly addressed these concerns in their revision, which is also missing markers to highlight changes w.r.t the original version.

In reason of this, while I do revise my score upwards, I still consider this paper to be below the acceptance threshold.

**Details Of Ethics Concerns:**

No concerns.

---

> ### Author Response · Authors · 2023-11-16
> **Response to reviewer YeEy [1/2]**
>
> We thank the reviewer for the detailed review and comments. Please see the following for our response!
>
> ### 1. The paper is written for a specialist audience, with little context or background given about NLP, about what the challenges in it are, about how ML methods can help solve it, or even just about what the tasks look like. In general, a formulation of the problem as a task that can be used to train a neural network on a certain experience and with a certain performance metric is omitted completely and taken for granted. As a result, most readers (as well as this referee) will not be able to get much from this paper.
>
> Our work is unrelated to NLP and the term “NLP” or “natural language processing” never exists throughout our paper. The focus of our work is on relational reasoning and decision-making tasks which is clearly stated many times in our paper as well as supplementary materials. Furthermore, it is a factual error to state that the description of the tasks and evaluation metrics are omitted. The corresponding details are all clearly delivered in section 4.1 and Appendix C with clear reference in the footnote of page 6. We sincerely recommend the reviewer carefully reread the paper and provide a more objective assessment.
>
> ### 2. My main recommendation would be for the authors to provide context on NLP, NLMs, and on how their task can be solved via ML methods to begin with, and how one would train a neural net to do so. Without such context, it's for me impossible to do justice to the content of this paper, but it is also going to be impossible for a mainstream ML reader (such as most of those attending ICLR) to profit from the paper, to begin with.
>
> Our work is unrelated to NLP. The concrete details of the NLM model are delivered in section 2.2 and Appendix D with clear reference in the footnote of page 4. The training details can be found in section 4.1 and Appendix B with clear reference on page 7. Again, We sincerely recommend the reviewer to carefully reread the paper and provide a more objective judgment.
>
> ### 3. On the contribution side, the authors should prove that their regularisation scheme does have any general applicability beyond NLP tasks carried out with NLMs.
>
> Our work is unrelated to NLP. The benchmarks evaluated in our work range from relational reasoning tasks and decision making tasks, which are all broadly used by many domains such as LLMs [1], program induction [2], SAT-based program synthesis [3], etc.
>
> ### 4. The authors' method appears to work as intended, but also to be quite narrow in applicability: it only presumably works on NLMs (since it requires direct control of the weights) and only for the particular task considered by the authors. At first sight, the scope of the contribution appears very limited. [1/2]
>
> It is a disparagement to regard our method as "work as intended" and "presumably works on NLMs and only for the particular task considered by the authors". First, we state that the domain of neural program induction is not narrow and limited and the NLM is an influential sota baseline: It manages to outperform previous enlighting methods such as DILP and MemNN. We also show that it also manages to outperform the sota gpt-4 model on the relation reasoning tasks. The results are shown in Table 1, where n denotes the number of family (graph) members.
>
> ||GPT-4||GPT-4 w/ CoT||NLM||
> |:--|:---|:---|:--:|:---:|:---|:---|
> |Task|n=20|n=30|n=20|n=30|n=20|n=30|
> |grandparents|93.97%|93.69%|95.67%|97.00%|96.15%|96.53%|
> |uncle|97.05%|96.44%|97.98%|97.87%|99.15%|98.68%|
> |mguncle|98.95%|99.00%|99.47%|99.46%|99.85%|99.68%|
> ||n=10|n=20|n=10|n=20|n=10|n=20|
> |1-outdegree|85.00%|76.50%|85.00%|76.50%|97.00%|85.50%|
> |6-connectivity|93.80%|58.58%|94.70%|58.58%|91.80%|91.73%|
>
> [**Table 1**: LLMs evaluation.]
>
> Specifically, all models are given samples with n=20 for few-shot learning, and tested with n=20 samples for performance and n=30 for generalization evaluation for the Family tasks. And for the graph reasoning tasks, all models are given samples with n=10 for few-shot learning, and tested with n=10 samples for performance and n=20 for generalization evaluation. The results indicate that the NLM outperforms the gpt-4 model as well as the gpt-4 with chain-of-shot mechanism [1] under few-shot learning setting with much smaller model size. It is also demonstrated in previous work that the LLM methods perform poorly on the blocksworld decision making benchmark [2] while NLM manages to excel it.

---

> ### Author Response · Authors · 2023-11-16
> **Response to reviewer YeEy [2/2]**
>
> ### 4.The authors' method appears to work as intended, but also to be quite narrow in applicability: it only presumably works on NLMs (since it requires direct control of the weights) and only for the particular task considered by the authors. At first sight, the scope of the contribution appears very limited. [2/2]
>
> To demonstrate the fact that FRGR is general, we also evaluated our method on more recent DLM model [3]. The results are shown in Table 2 & 3. The results indicate that FRGR is also effective in improving the DLM model under both the data-rich and data-scarce settings.
>
> |Family Tree|DLM||||DLM w/ FRGR (Ours)||||
> |:--:|:---:|:--:|:--:|:--:|:--:|:--:|:--:|:--:|
> ||\# Iterations|Grad-ratio|n=20|n=100|\# Iterations|Grad-ratio|n=20|n=100|
> |HasFather|22.00(±4.69)|100.00%|100.00%(±0.00)|100.00%(±0.00)|23.6(±7.13)|100.00%|100.00%(±0.00)|100.00%(±0.00)|
> |HasSister|68.80(±12.67)|100.00%|100.00%(±0.00)|100.00%(±0.00)|67.20(±3.27)|100.00%|100.00%(±0.00)|100.00%(±0.00)|
> |IsGrandparent|50.80(±14.46)|100.00%|100.00%(±0.00)|100.00%(±0.00)|55.60(±7.40)|100.00%|100.00%(±0.00)|100.00%(±0.00)|
> |IsUncle|319.20(±165.54)|60.00%|60.00%(±0.49)|60.00%(±0.49)|278.40(±155.48)|80.00%|80.00%(±0.40)|80.00%(±0.40)|
> |IsMGUncle|459.20(±91.23)|40.00%|48.1%(±0.30)|20.00%(±0.40)|423.80(±104.35)|60.00%|58.20%(±0.35)|40.00%(±0.49)|
> |Graph|DLM||||DLM w/ FRGR (Ours)||||
> ||\# Iterations|Grad-ratio|n=10|n=20|\# Iterations|Grad-ratio|n=10|n=20|
> |1-OutDegree|46.20(±2.39)|100.00%|100.00%(±0.00)|100.00%(±0.00)|50.00(±7.24)|100.00%|100.00%(±0.00)|100.00%(±0.00)|
> |2-OutDegree|81.60(±13.74)|100.00%|100.00%(±0.10)|100.00%(±0.28)|73.60(±3.84)|100.00%|100.00%(±0.00)|100.00%(±0.00)|
> |4-Connectivity|90.40(±22.42)|100.00%|100.00%(±0.00)|100.00%(±0.00)|87.40(±15.09)|100.00%|100.00%(±0.00)|100.00%(±0.00)|
> |6-Connectivity|282.40(±146.44)|80.00%|86.90%(±0.26)|53.28%(±0.34)|230.80(±166.14)|80.00%|95.40%(±0.09)|90.10%(±0.20)|
>
> [**Table 2**: Results under data-rich scenario with DLM and DLM-FRGR]
>
> |Family Tree|DLM||||DLM w/ FRGR (Ours)||||
> |:--:|:---:|:--:|:--:|:--:|:--:|:--:|:--:|:--:|
> ||\# Iterations|Grad-ratio|n=20|n=100|\# Iterations|Grad-ratio|n=20|n=100|
> |HasFather|23.20(±3.63)|100.00%|100.00%(±0.00)|100.00%(±0.00)|27.00(±6.41)|100.00%|100.00%(±0.00)|100.00%(±0.00)|
> |HasSister|58.80(±7.56)|100.00%|100.00%(±0.00)|100.00%(±0.00)|67.60(±7.67)|100.00%|100.00%(±0.00)|100.00%(±0.00)|
> |IsGrandparent|44.40(±4.77)|100.00%|100.00%(±0.00)|100.00%(±0.00)|56.80(±7.82)|100.00%|100.00%(±0.00)|100.00%(±0.00)|
> |IsUncle|401.60(±149.62)|40.00%|40.20%(±0.49)|40.00%(±0.49)|362.80(±141.94)|80.00%|80.00%(±0.40)|80.00%(±0.40)|
> |IsMGUncle|500.00(±0.00)|0.00%|0.00%(±0.00)|0.00%(±0.00)|500.00(±0.00)|0.00%|0.00%(±0.00)|0.00%(±0.00)|
> |Graph|DLM||||DLM w/ FRGR (Ours)||||
> ||\# Iterations|Grad-ratio|n=10|n=20|\# Iterations|Grad-ratio|n=10|n=20|
> |1-OutDegree|47.20(±2.28)|100.00%|100.00%(±0.00)|100.00%(±0.00)|48.00(±5.83)|100.00%|100.00%(±0.00)|100.00%(±0.00)|
> |2-OutDegree|92.00(±36.93)|100.00%|100.00%(±0.10)|100.00%(±0.28)|83.620(±27.69)|100.00%|100.00%(±0.00)|100.00%(±0.00)|
> |4-Connectivity|82.80(±15.27)|100.00%|100.00%(±0.00)|100.00%(±0.00)|68.00(±5.83)|100.00%|100.00%(±0.00)|100.00%(±0.00)|
> |6-Connectivity|424.00(±169.94)|20.00%|75.30%(±0.22)|59.80%(±0.23)|359.20(±196.01)|40.00%|83.60%(±0.17)|70.00%(±0.261)|
>
> [**Table 3**: Results under data-scarce scenario with DLM and DLM-FRGR]
>
> Second, in terms of the tasks used for evaluation in our work, it is inappropriate to state the "...only for the particular task considered by the authors" as we did not cherry-pick the evaluation tasks. All the benchmarks we used are directly adopted from the NLM and DLM papers. All these tasks are also broadly used for the reasoning ability evaluation of LLMs [2] as well as SAT-based program synthesis [4].
> In summary, all the above-mentioned evidence suggests that FRGR is not "intended" and "limited in contribution" by significantly improving over the state-of-the-art neural program induction models (which are sota in terms of relational reasoning ability) on previously adopted and broadly used benchmarks.
>
> [1] Wei, Jason, et al. "Chain-of-thought prompting elicits reasoning in large language models." Advances in Neural Information Processing Systems 35 (2022): 24824-24837.
>
> [2] Valmeekam, Karthik, et al. "Large Language Models Still Can't Plan (A Benchmark for LLMs on Planning and Reasoning about Change)." NeurIPS 2022 Foundation Models for Decision Making Workshop. 2022.
>
> [3] Matthieu Zimmer, et al. "Differentiable Logic Machines". Transactions on Machine Learning Research. (2023).
>
> [4] Raghothaman, Mukund, et al. "Provenance-guided synthesis of datalog programs." Proceedings of the ACM on Programming Languages 4.POPL (2019): 1-27.

---

> ### Author Response · Authors · 2023-11-20
>
> We would appreciate it if the reviewer can be more involved in the discussion (Nov 10-22) and let us know whether the response has clarified the corresponding questions.

---

> > ### Comment · Reviewer_JknR · 2023-11-20
> >
> > By NLP the review may have meant neural logic programs, or neural logic programming. given the other meaning of NLP, this is confusing either way.

---

> > ### Comment · Reviewer_YeEy · 2023-11-21
> >
> > I thank the authors for their clarifications on the paper and I apologize for not engaging earlier.
> >
> > First of all, yes, with NLP I meant Neural Logic Programming, not Neural Linguistic Programming. I thank referee JknR for suggesting that this might be the case. It is true, the acronym NLP never appears in the paper, but I though that what I was referring was obvious from the context. At any rate, the authors could have simply asked for clarification rather than jumping to conclusions, especially considering that their belief that I had mistaken their work for relating to Neural Linguistic Programming is featured in three of the points in their reply.
> >
> > I get the impression that the authors misconstrued my review as an attack to their work. I wish to point out that this is not the case and that I myself stated that my confidence in the score I assigned is low. To be more concrete, when I stated that their method works "as intended" I was not implying that they have merely found a solution for a problem they invented.
> >
> > I was under the mistaken impression that NLM and Neural Logic Programming were "tasks" whilst, after re-reading the paper and the authors' reply, I can now see that they are fundamentally architectures/inductive biases which can indeed be applied to multiple tasks. I retract my statement that the authors only considered one particular task with limited impact in order to test their method. It is indeed true that they apply their method to multiple and relevant benchmarks, and therefore their work has indeed more relevance and applicability than I first claimed.
> >
> > Still, my concerns on presentation and writing remained mostly unaddressed. I re-read the paper, and the writing of section 2 and 3 still seems as unclear as it was before the revision. As a matter of fact, the revised version does not clearly highlight which changes, if any, were made to the paper. The authors' responded to my concerns on their writing by simply requiring that I read the paper again; all the other referees raised similar concerns, and it does not seem to me that the authors addressed them much beyond providing explanations in their replies, whilst they should have revised the writing in the paper itself. I cannot speak for the other reviewers, but when I claim that a paper is difficult to read, I am more concerned about its potential future readers than myself. While my initial underestimation of the relevance of NLMs (and by extension, of the authors' work) to broader ML research is definitely due to my own self-admitted ignorance of that field, maybe a more carefully written introduction and sections 2 and 3 would have considerably helped me and any readers in putting the authors' work in the proper context.
> >
> > Based on this, I shall revise my score upward to 5 and my confidence in this assessment to 3.

---

> ### Author Response · Authors · 2023-11-21
>
> We would appreciate it if the reviewer could let us know whether the response has clarified the corresponding questions as it is approaching the Author/Reviewer Discussion period deadline (Nov 22) and it has already been 5 days since we posted our response (Nov 16).

---

> ### Author Response · Authors · 2023-11-21
> **revised version**
>
> Thanks for the response. We have not submitted the revised version of the paper before as we think the reviewer has not carefully read the paper due to the misleading use of "NLP" as well as the previous unfair claims. We have now submitted the revised version  of the main paper as well as the supplementary materials which explain NLM model design more plainly in the preliminary and we use a concrete case to explain the rationale of FRGR in the methodology section. Besides, we have revised all the mentioned details by all the reviewers as well. The revised parts are highlighted in blue. We hope the reviewer to inspect our revised version and take it into consideration for the rating.

---

### Official Review · Reviewer_JknR · 2023-10-31

**Soundness:** 4 excellent
**Presentation:** 3 good
**Contribution:** 4 excellent
**Rating:** 8
**Confidence:** 4

**Summary:**

This paper proposes a training modification to improve the training efficiency, generalization, and performance of neural models for inductive logic programming. Motivated by provenance-guided SAT-based synthesis techniques, the proposed technique stores an ongoing list which contains provenance information for each error experienced, in the form of the weight location that most contributed to an error.
Then it applies a pattern mining algorithm to summarize the errors, and applies L1 regularization to the erroneous neurons during training. Experiments show that with this training modification, the existing neural ILP approach of Neural Logic Machines improves in performance, data efficiency, computational efficiency, and generalization.

**Strengths:**

- The technique is quite novel, at least to me. I have never seen a technique of attempting to improve data efficiency of neural network training by cataloguing errors and penalizing their recurrence with L1 regularization. The success of a similar technique in SAT-based program synthesis motivates the technique well, and it appears this works due to the close connection between the NN architecture and the logical meaning of weight activations.
- The improvements in performance seem significant! Performance is never decreased, and especially large improvements are seen in complex tasks and in data-scarce settings.
- The paper is well-written.
- The approach is simple and general enough that I would guess it has a good chance of successfully apply to other settings.
- The experiments seem thorough: they cover both supervised and RL tasks, the same tasks covered in prior work, and additionally evaluate in both normal and data-scarce settings.

**Weaknesses:**

There are two main weaknesses to the paper: (1) there is not much attempt to understand why the technique works. (2) some of the writing in section 2 and 3 is not clear or fully explained. (1) is a much more important weakness, as (2) can be fixed pretty easily.

(1) there is not much attempt to understand why the technique works.
- Given how novel the approach is to me, I have a hard time forming a mental model connecting technique's description with the resulting performance.
    - No ablations or comparisons of design choices, or even motivation given for the design choices.
    - The paper would be stronger with experiments (could be toy) or examples that show how the approach is working, or a discussion comparing the usefulness of provenance information for SAT-based synthesis with how the provenance information is helping the neural models
    - I think there should be more description in section 3 of why the technique works: I guess the model is  a bunch of logical combinations simultaneously during training, and due to the logical nature of the NN activations, it's possible to get information about "incorrect search" happening just how the SAT solver generates erroneous formula.
   - see questions section for more questions attempting to understand and contextualize the technique.

(2) unclear writing
- The description of NLM's in section 2.2 is not super clear. I'm unsure how much it can be improved while keeping the section short, but I would recommend working on it. In particular, the original NLM paper did a much better job by intuitively summarizing the approach before describing the details more formally.
- The description of the approach in section 3 is poor. it leaves out numerous details, or mentions them unsystematically. things like "what is an error?", eq 1 is challenging to understand, the error list updating is never described clearly outside of the pseudocode— based on the pseudocode, you store a list of the T most recent errors, and once T are found, you run the pattern mining algorithm. and then you start over with a new list, right? this could be quickly described with a sentence in the text.
- what does the Apriori algorithm return? a single thing? based off elsewhere, it returns a set of weights, but this should be described when Apriori is brought up.
- Supplementary material could do a better job of describing the inputs and outputs of the model for the relational reasoning tasks and the rRL tasks.

To summarize, I have no criticisms of the technique, experiments, or results, but **I do wish there were more justification of the design choices, ablations, other experiments, discussion, or comparison to related techniques, to help understand why this approach works**.

**Questions:**

- How much of a correspondence is there between the form of the provenance information in this paper and RAGHOTHAMAN et al 2019?
-  RAGHOTHAMAN et al 2019 provides guidance for both "why" an incorrect conclusion was derived and "why not" i.e. why a correct conclusion was not derived, but this work only provides guidance for "why". what about "why not" mistakes?
    - To elaborate more, regularization of errors is never done with NN training in any other domain. Does this approach fail when applied to image classification? Both a yes or no answer would be quite illuminating...
- Is there any related work of such types of techniques? (l1 penalty on negative examples, storing errors during training)

Answering these questions as well as those raised in the Weaknesses section regarding the lack of attempt to understand why the method works would improve my opinion of the paper.

---

> ### Author Response · Authors · 2023-11-16
> **Response to reviewer JknR [1/2]**
>
> We thank the reviewer for the detailed review and comments. Please see the following for our response!
> ### 1. No ablations or comparisons of design choices, or even motivation given for the design choices.
>
> We conduct an ablation study by comparing our approach with a random regularization baseline (i.e., randomly penalizing same percentage of the model’s weights as that of FRGR), denoted as NLM w/ random. The results are shown in Table 1:
>
> |2-Outdegre | grad-rate |n=10| n=20| Iterations |
> | :--| :--| :-- | :--- | :--- |
> | NLM| 90.00% | 96.52%±0.10  |  90\.80%±0.28  | 77\.9±150.5  |
> | NLM w/ random | 80.00%   | 82\.10%±0.36  |  81\.18%±0.38  | 114±203.70  |
> | NLM w/ FRGR | 100.00%| 100.00%±0.00 |100\.00%±0.00 | 13\.40±11.13 |
>
> [**Table 1**: regularization comparisons.]
>
> The results illustrate that by adding random regularization, the model's performance decreases compared to that of the original NLM. Specifically, compared to the original NLM model, the grad-ratio decreases by +10.00%, and the performance decreases by -14.42%.
> While with FRGR, it is effective in improving grad-ratio, performance, generalization, and training efficiency.
> The results demonstrate that the vanilla training loss, random regularization as well as weight decay regularization are not sufficient enough to capture the repetitious error pattern during training thus incapable of improving the NLM model.
>
> ### 2. The paper would be stronger with experiments (could be toy) or examples that show how the approach is working, or a discussion comparing the usefulness of provenance information for SAT-based synthesis with how the provenance information is helping the neural models
>
> We present an error pattern that FRGR manages to mitigate. Concretely, for the IsMGUncle and IsUncle tasks, their ground-truth programs only involve using binary predicates.  The ground-truth programs are shown as follows:
>
> IsUncle:
>
> $$
>  \operatorname{IsUncle}(X, Y) \leftarrow \exists Z,  ((\operatorname{IsMother}(X, Z) \wedge \operatorname{IsBrother}(Z, Y))) \\
>  \vee(\operatorname{IsFather}(X, Z) \wedge \operatorname{IsBrother}(Z, Y)) \\
> $$
> $$
> \operatorname{IsBrother}(X, Y) \leftarrow \exists Z,  ((\operatorname{IsSon}(Z, Y) \wedge \operatorname{IsSon}(Z, X)) \\
>  \vee(\operatorname{IsSon}(Z, Y) \wedge \operatorname{IsDaughter}(Z, X)))
> $$
>
> IsMGUncle:
> $$
> \operatorname{IsMGUncle}(X, Y) \leftarrow \exists Z,(\operatorname{IsMother}(X, Z) \wedge \operatorname{Is} \operatorname{Uncle}(Z, Y))
> $$
> Therefore, it is considered an erroneous behavior if the model strongly attributes its induction based on unary predicates. We calculate the ratio of all output binary predicates whose most attributed input predicate is unary during the training process. We evenly sample 5 epochs from the training process till NLM w/ FRGR converges to an optimal solution. The results are shown in Table 2. It is obvious that with the use of FRGR, the ratio of unary predicates is rapidly decreased compared with the vanilla NLM. The results indicate the FRGR is effective in recognizing repetitious error patterns during the training process and actively regularizing the model based on it, which results in higher training efficiency and better generalization.
>
> || **IsMGUncle** || |  |  |
> | :-- | :-- | :-- | :-- | :-- | :- |
> | NLM  |0\.417 | 0\.444 | 0\.426 | 0\.486 | 0\.394 |
> | NLM w/ FRGR | 0\.417| 0\.426 | 0\.454 | 0\.394 | 0\.356 |
> | | **IsUncle** || | | |
> |NLM | 0\.433 | 0\.458 | 0\.450 | 0\.433 | 0\.483 |
> |NLM w/ FRGR  |0\.433| 0\.442 | 0\.467 | 0\.358 | 0\.375 |
>
> [**Table 2**: interpretation analysis.]
>
> This process is similar to that of the provenance information used in the SAT-based synthesis, in which the solver's error is actively summarized based on previous derivation trees and is used to reduce future search space.
>
> ### 3. I think there should be more description in section 3 of why the technique works: I guess the model is a bunch of logical combinations simultaneously during training, and due to the logical nature of the NN activations, it's possible to get information about "incorrect search" happening just how the SAT solver generates erroneous formula.
>
> We agree with the explanation of the reviewer that due to the similarity of the neuro-symbolic models' design and the SAT solving process, we can therefore effectively leverage the error pattern of the model's previous learning process (similar to the provenance information summarized from derivation tree which is used in SAT-based synthesis) to improve models' generalization and training efficiency. We will add the corresponding explanation of our method's effectiveness in the paper.

---

> > ### Comment · Reviewer_JknR · 2023-11-20
> >
> > Thank you for answering my questions. It helps me understand the effectiveness of this approach better. Cool! I have revised my score from a 6 to an 8.

---

> > > ### Author Response · Authors · 2023-11-20
> > >
> > > We sincerely thank the reviewer for the recognition of our work!

---

> ### Author Response · Authors · 2023-11-16
> **Response to reviewer JknR [2/2]**
>
> ### 4. The description of the approach in section 3 is poor. it leaves out numerous details, or mentions them unsystematically. things like "what is an error?", eq 1 is challenging to understand, the error list updating is never described clearly outside of the pseudocode— based on the pseudocode, you store a list of the T most recent errors, and once T is found, you run the pattern mining algorithm. and then you start over with a new list, right? this could be quickly described with a sentence in the text.
>
> An error simply denotes a wrong final conclusion (i.e., classification results for relational reasoning tasks, actions for decision-making tasks). The understanding of the reviewer in terms of the error list update process is mostly correct: we store a list of the T most recent erroneous behavior in the erroneous behavior list, and once T is found, we run the pattern mining algorithm. And we continuously update the list from the beginning of the erroneous behavior list. We will add more explanation of the update process in our paper.
>
> ### 5. What does the Apriori algorithm return? a single thing? based off elsewhere, it returns a set of weights, but this should be described when Apriori is brought up.
>
> Apriori returns the position index of the frequently occurred weights of the error behavior representations stored in the erroneous behavior list, as described with a concrete example after Apriori brought up (last paragraph of section 3.1): “After the mining process, FRGR manages to identify the error pattern representation that stores the position index of the frequently intersected weights of all historical erroneous behavior representations, …”
>
> ### 6. How much of a correspondence is there between the form of the provenance information in this paper and RAGHOTHAMAN et al 2019?
>
> FRGR is motivated by and similar to the provenance information used in SAT-based synthesis, which leverages error from model's previous derivation to reduce future search space. FRGR is the first to propose actively summarizing neural model's error pattern from previous mistakes and using it to boost the model's training efficiency and generalization. The corresponding details are illustrated in the analogy between derivation tree based provenance used in SAT-based synthesis and the neuro-symbolic model architectures shown in the Introduction (section 1).
>
> ### 7. RAGHOTHAMAN et al 2019 provides guidance for both "why" an incorrect conclusion was derived and "why not" i.e. why a correct conclusion was not derived, but this work only provides guidance for "why". what about "why not" mistakes?
>
> In this paper, we focus on the realization of the “why” provenance on the neural program induction model. This idea of further using “why not” for neural program induction is intriguing and would be explored in our future work.
>
>
> ### 8. To elaborate more, regularization of errors is never done with NN training in any other domain. Does this approach fail when applied to image classification? Both a yes or no answer would be quite illuminating...
>
> Though in this work, we focus our scope on relational reasoning and decision-making tasks, we believe that our method is general and can be well adapted to other domains such as image classification, robotics, etc. We will strive to further apply our methods to other areas if our paper gets accepted.
>
> ### 9. Is there any related work of such types of techniques? (L1 penalty on negative examples, storing errors during training)
>
> To the best of our knowledge, there are no similar techniques and we are the first to conduct pattern mining on neural model’s previous erroneous behavior and use it to improve training efficiency and generalization.

---

> ### Author Response · Authors · 2023-11-20
>
> We would appreciate it if the reviewer can be more involved in the discussion (Nov 10-22) and let us know whether the response has clarified the corresponding questions.

---

### Official Review · Reviewer_7KpG · 2023-11-01

**Soundness:** 3 good
**Presentation:** 3 good
**Contribution:** 2 fair
**Rating:** 5
**Confidence:** 3

**Summary:**

This work proposes a regularization technique called FRGR that leverages a buffer of common recent mistakes during training. The method is implemented on top of neural logic machines (NLM), a neurosymbolic architecture for logic programs. Over two domains (logical reasoning and decision-making), FRGR shows improvements in performance over a vanilla NLM baseline both in the standard setting as well as a low-data regime.

**Strengths:**

The central idea to augment the training process using representations of past failures is interesting and fairly general, and could be applicable to many important domains.

**Weaknesses:**

The paper is unclear on a number of expository details. $\theta_\nu$ in Equation (2) should be given an explicit formula, as this is a key term in the regularizer. The notation is overloaded, for instance $m$ refers to both the size of $\mathcal{U}$ in Section 2.2 as well as the matching factor in Section 3.2. I also could not find how # of iterations is defined in either the main paper or the appendix, despite this being one of the claimed improvements of FRGR over vanilla NLM.

I also have some concerns about the experimental results. First, many choices of hyperparameters are not explored, such as the choice of the buffer size $\tau$. It would be good to perform some ablation studies exploring the effect of this hyperparameter. As there is no theory to justify the regularizer, I think this raises the bar for empirical results. Second, the improvement in # of iterations appears not to be statistically significant, as the errors appear quite large. The improvements in Figure 4 appear quite marginal, again with overlapping error bars.

Finally, the related work is missing a number of relevant lines of research, including hard negative mining ([1], i.a.) and experience replay ([2], i.a.).

[1] Feng, Yu, et al. "Program synthesis using conflict-driven learning." ACM SIGPLAN Notices 53.4 (2018): 420-435.

[2] Continuous control with deep reinforcement learning. Timothy P. Lillicrap, Jonathan J. Hunt, Alexander Pritzel, Nicolas Heess, Tom Erez, Yuval Tassa, David Silver, Daan Wierstra. ICLR 2016.

**Questions:**

In Tables 1 and 2, what are units for the reported errors? Specifically, the entries are generally of the form x% +/- y, so is y a raw number or a percentage?

How is $\theta_\nu$ defined?

How did you determine when to stop training (for # of iterations)?

===

Post rebuttal I have increased my scores for soundness and presentation, as well as my rating from 3 to 5.

---

> ### Author Response · Authors · 2023-11-16
> **Response to reviewer 7KpG**
>
> We thank the reviewer for the detailed review and comments. Please see the following for our response!
>
> ### 1. I also could not find how # of iterations is defined in either the main paper or the appendix
>
> The concept of iterations is defined in the Evaluation Metrics section of page 7. Training iteration measures the number of training epochs required to train the model till reaching optimal success rate on validation set. We use it to measure training efficiency.
>
> ### 2. the choice of the buffer size $\tau$
>
> The default value of $\tau$ is set to 100. We conduct ablation studies to explore the effect of $\tau$. Specifically, we conduct experiments on the IsUncle task under three different values of $\tau$: 75, 100, 125, the results are shown in Table 1 & 2:
>
> | tau | grad-rate | n=20       | n=100      | \# iterations |
> | :-- | :-- | :--- | :--- | :--- |
> | 75  | 100.00%  | 100%±0.00 | 100%±0.00 | 83±22.75     |
> | 100 | 100.00%  | 100%±0.00 | 100%±0.00 | 82.50±17.60  |
> | 125 | 100.00%  | 100%±0.00 | 100%±0.00 | 96.50±25.61  |
>
> [**Table 1**: erroneous behavior list size analysis on IsUncle.]
>
> | tau | grad-rate | n=10         | n=20         | \# iterations  |
> | :-- | :-- | :----- | :----- | :---- |
> | 75  | 80.00%   | 90.22%±0.18 | 84.44%±0.29 | 143.6±188.76  |
> | 100 | 90.00%   | 97.12%±0.09 | 94.60%±0.16 | 69.2±153.32   |
> | 125 | 90.00%   | 95.33%±0.13 | 91.78%±0.23 | 117.00±143.41 |
>
> [**Table 2**: erroneous behavior list size analysis on 6-connectivity.]
>
> The results indicate that under all the evaluated settings of $\tau$, FRGR steadily improves over the original NLM model.
>
> ### 3. Second, the improvement in # of iterations appears not to be statistically significant, as the errors appear quite large.
>
> The reason for the large errors (standard deviation) is that our model manages to converge to the optimal solution for most of the seeds, therefore a single timeout (reaching the maximum 500 epochs) would render the standard deviation of the results large. Whereas the original NLM model often fails to converge to an optimal solution and therefore frequently timeout. For example, Table 3 shows the results of all the evaluated seeds of the 6-Connectivity task (note that 500 training epochs is the timeout threshold):
>
> | NLM      | 151     | 500    | 10     | 500    | 500    | 176    | 500    | 500    | 62     | 500    |
> | :---- | :--- | :-- | :-- | :-- | :-- | :-- | :-- | :-- | :-- | :-- |
> | **Ours** | **500** | **12** | **10** | **10** | **15** | **90** | **15** | **12** | **11** | **17** |
>
> [**Table 3**: detailed training epochs of 6-Connectivity.]
>
> ### 4. The improvements in Figure 4 appear quite marginal, again with overlapping error bars.
>
> We state that the improvement is not marginal under the data-scare setting if the reviewer is referring to the curves of performance and generalization. Though the lines of the NLM and NLM w/ frgr could be close when the data volume grows larger (due to the scale of the image), we have to mention that it is critical and also difficult for the models to reach an optimal solution (100\%) (i.e., inducing the perfect ground-truth program), and with the incorporation of FRGR, the model manages to achieve optimal performance with much fewer iterations. For instance,  NLM w/ FRGR manages to achieve optimal solution with only 800 examples on the IsUncle task, whereas the original NLM never able to achieve that across multiple seeds (only ~98%) with even 1000 examples. The results is consistent with standard deviation = 0 across 10 random seeds. Similarly, for outdegree-2, NLM w/ FRGR achieves optimal performance and generalization with only 200 examples, while the original NLM takes 600 examples.
>
> ### 5. In Tables 1 and 2, what are units for the reported errors? Specifically, the entries are generally of the form x% +/- y, so is y a raw number or a percentage?
>
> y is a raw number. We will add the description regarding this format in our paper.
>
> ### 6. How is $\theta_v$ defined?
>
> $\theta_v$ denotes the the set of instersected weights of the error pattern set and the model’s current behavior representation, which is described in the last paragraph of section 3.2: “In this work, we consider matching by calculating the intersection between the error pattern set $\epsilon$ and the model’s current behavior representation $\omega$, as shown in line 13. This matching factor, denoted by $m$, measures the degree of repetitions (regarding the error pattern presented previously) performed by the model.”, “To calculate the behavioral regularization term, we obtain the intersected weights set m and apply the L1 norm to aggregate the weights …”
>
> ### 7. How did you determine when to stop training (for # of iterations)?
>
> The training stops when the model achieves 100% success rate on the validation set. The maximum number of epochs is set as 500 for all models, which strictly follows the settings of the original NLM paper.

---

> > ### Comment · Reviewer_7KpG · 2023-11-21
> >
> > > Training iteration measures the number of training epochs required to train the model till reaching optimal success rate on validation set
> >
> > Thanks for clarifying. I do not find this clear from the current description in Section 7 ("Finally, training iteration refers to the number of iterations used for training.") and would suggest replacing the text in the versions.
> >
> > > The reason for the large errors... a single timeout
> >
> > Ok, but this just suggests that the standard deviation is not an informative measure of statistical significance for your data. Consider using something like quantile statistics instead
> >
> > > \theta_\nu denotes the the set of instersected weights of the error pattern set and the model’s current behavior representation
> >
> > Perhaps this is just because I am unfamiliar with ILP, but this notation is quite confusing. $\theta$ refers to the parameters of the MLP but $\theta_\nu$ refers to a set of intersected weights. I'll reiterate my request that the revision have an explicit formula for $\theta_\nu$.

---

> > > ### Author Response · Authors · 2023-11-21
> > >
> > > Thanks for the response. We have now submitted the revised version of the main paper as well as the supplementary materials which updated all the mentioned details by the reviewer. We hope the reviewer to inspect our revised version and take it into consideration for the rating.

---

> > > > ### Comment · Reviewer_7KpG · 2023-11-22
> > > >
> > > > Thanks. I have read the revised version. I find the new notation $\theta[\nu]$ to be more clear, and the background on NLM is great as well.
> > > >
> > > > Having said that, I still struggle to recommend the paper for acceptance. Right now, the paper proposes a regularizer on top of NLM, and the empirical results demonstrate an improvement over that single base architecture. I'm not very familiar with this subfield, but I think it would greatly improve the contribution to increase the scope of the comparisons. Perhaps with more discussion this could have been resolved, but unfortunately the paper suffered from major clarity issues until very late in the discussion period.
> > > >
> > > > I would also still recommend the authors use a different presentation for their error bars in a future revision.

---

> ### Author Response · Authors · 2023-11-20
>
> We would appreciate it if the reviewer can be more involved in the discussion (Nov 10-22) and let us know whether the response has clarified the corresponding questions.

---

> > ### Author Response · Authors · 2023-11-21
> >
> > We would appreciate it if the reviewer could let us know whether the response has clarified the corresponding questions as it is approaching the Author/Reviewer Discussion period deadline (Nov 22) and it has already been 5 days since we posted our response (Nov 16).

---

### Meta-Review · Area_Chair_QyUk · 2023-12-04

**Metareview:**

This paper introduces a new method for Inductive Logic Programming (ILP). It blends classic symbolic methods with differentiable relaxations of first-order logic, building primarily upon Neural Logic Machines.

The primary advantage of differentiable fuzzy approaches to ILP is that (1) they can handle noise in the training data and (2) they can interoperate with neural networks. However, none of the experiments in this paper examine either noisy training data, or interoperation with conventional neural networks (for an example ILP system that does both, see the cited Evans & Grefenstette '18).

Therefore, the work considers purely symbolic ILP problems. But it does not compare with any modern ILP systems, such as Popper or Metagol. If the goal is to do neurosymbolic ILP, then the experiments need to include either noise or interoperation with neural networks. If the goal is to do conventional ILP, then conventional ILP must be compared with.
Unfortunately, the proposed method is quite complicated, with many moving pieces, so it is unlikely to have impact purely on the algorithm -- it needs the right experiments to convince the reader that this algorithm is "worth it," but such experiments are not there.

Reviewers are on balance mostly in favor of rejecting the paper, although for slightly different reasons (eg, lack of clarity in the manuscript itself). Although many of these criticisms are valid, in my opinion, the key problem with this paper is that it presents a very complicated algorithm that does not provide the key experiments and comparisons needed to motivate it.

**Justification For Why Not Higher Score:**

The paper presents a neurosymbolic ILP system but only evaluates on symbolic tasks, and does not compare against modern symbolic ILP systems (which would almost certainly do better on those benchmarks). It must either compare on neurosymbolic problems or show advantages against symbolic systems on symbolic problems.

**Justification For Why Not Lower Score:**

n/a

---

### Decision · Program_Chairs · 2024-01-16

Reject